# Proinflammatory polarization of engineered heat-inducible macrophages reprogram the tumor immune microenvironment during cancer immunotherapy

Yanan Xue[1,2,5], Xiaojie Yan[2,3,5], Da Li[1,5], Shurong Dong[4] & Yuan Ping ●[1,2,3] ✉

The success of macrophage-based adoptive cell therapy is largely constrained by poor polarization from alternatively activated (M2-like) to classically activated (M1-like) phenotype in the immunosuppressive tumor microenvironment (TME). Here, we show that the engineered macrophage (eMac) with a heat-inducible genetic switch can induce both self-polarization of adoptively transferred eMac and re-polarization of tumour-associated macrophages in response to mild temperature elevation in a mouse model. The locoregional production of proinflammatory cytokines by eMac in the TME dose not only induces the strong polarization of macrophages into a classically activated phenotype, but also ensures that the side effects typical for systemically administrate proinflammatory cytokines are avoided. We also present a wearable warming device which is adaptable for human patients and can be remotely controlled by a smartphone. In summary, our work represents a safe and efficient adoptive transfer immunotherapy method with potential for human translation.

Adoptive cell therapy is recently emerging as one of the most viable clinical approaches for cancer immunotherapy[1–4]. Among various adoptive cell therapies, chimeric antigen receptor (CAR) T-cell therapy, in which T cells are engineered to express CARs, has shown great potential for treating cancer patients with B cell malignancies[5]. Despite the clinical advances in hematological malignancies, therapeutic efficacy of CAR-T-cell therapy against solid tumors remains moderate, largely owing to the limited infiltration into the tumor tissue and the inhibited activation in the tumor microenvironment (TME)[6]. In addition, serious side effects including cytokine release syndrome and neurotoxicity are also of the major concerns for clinical CAR-T-cell therapy[7]. Recently, macrophage-based cell therapies show promising results in overcoming several critical challenges that CAR-T-cell therapy encounters in treating solid tumors and represents a new paradigm for cancer immunotherapy[3]. As innate immune cells, macrophages are plastic and possess unique abilities to polarize toward different phenotypes, which is tightly regulated by their surrounding microenvironment[8]. Tumors recruit circulating monocytes and tissue-resident macrophages to the TME, where they are polarized toward an M2-like phenotype to constitute tumor-associated macrophages (TAMs) and are involved in tumor progression, immunosuppression and metastasis[9,10]. In contrast, the polarized M2 macrophages can be re-polarized into M1 phenotype in response to external stimuli like cytokines and become proinflammatory and antitumoral by inducing phagocytosis, producing copious amounts of proinflammatory cytokines and activating cytotoxic T lymphocytes[3,9].

To this end, notable efforts have been dedicated to such a re-polarization ex vivo or in vivo to enhance the immunotherapeutic potency of macrophages[11–15]. Of these endeavors, interferon-gamma (IFN-γ), as an important class of proinflammatory cytokine, is often

[1]Sir Run Run Shaw Hospital, School of Medicine, Zhejiang University, Hangzhou 310016, China. [2]College of Pharmaceutical Sciences, Zhejiang University, Hangzhou 310058, China. [3]Liangzhu Laboratory, Zhejiang University, Hangzhou 311121, China. [4]College of Information Science and Electronic Engineering, Zhejiang University, Hangzhou 310027, China. [5]These authors contributed equally: Yanan Xue, Xiaojie Yan, Da Li. ✉e-mail: pingy@zju.edu.cn

exploited to repolarize macrophages into M1 phenotype[9,16]. However, IFN-γ is rapidly cleared from the blood upon systemic administration, and the frequent re-administration of IFN-γ is therefore essential to ensure the sufficient locoregional concentration that is required for M2 to M1 polarization, often leading to the systemic toxicity and side effects, such as fever, diarrhea, or even neurotoxicity[17–19]. Several recent investigations also indicated that high serum concentration of IFN-γ may also play a pro-tumorigenic role by downregulating major histocompatibility complexes and upregulating checkpoint inhibitors (such as programmed cell death ligand 1)[19–21]. Recently, Mitragotri and co-workers reported that the polarization toward antitumor M1 phenotype can be continuously induced in vivo by means of backpacking IFN-γ-loaded discoidal particles onto the macrophage surface, which well avoid unwanted reversible polarization of M1 to M2 and initiate strong antitumor effect in a murine breast cancer model with strong immunosuppression[15]. Though the backpacked IFN-γ could continuously guide the polarization of macrophages toward M1 phenotype, these macrophages are limited to the intratumoral injection, possibly due to the systemic leaky effect of IFN-γ from the discoidal particles. Thus, the essential locoregional concentration of IFN-γ in the TME is critical to direct the successful re-polarization as well as to address the adverse effects of IFN-γ.

Herein, we develop an engineered macrophage (eMac) which can locoregionally secrete IFN-γ in the tumor tissue through wireless remote control to induce their polarization toward M1 phenotype for cancer immunotherapy (Fig. 1). Of note, by virtue of HSP70-based heat shock promoter, the production of IFN-γ from eMac is precisely regulated by a wirelessly controlled, intelligent wearable warming device (iWarm) that spatiotemporally controls the endogenous gene activation and deactivation to avoid the unwanted side effects. The remotely controlled polarization of eMac through a user-friendly, intelligent device is expected to become a personalized, precision medicine for adoptive cell therapy.

## Results

### Characterization of the iWarm-controlled gene expression mediated by heat shock promoter

We designed an intelligent warming device which consists of two parts: a smart device where the interface of application (APP) for the remote control of iWarm is user-friendly (Fig. 2a), and a piece of customized, wearable garment by which iWarm can be incorporated for locoregional heating. Key components of the heating device include: (1) an organic light emitting diode (OLED) display panel displaying real-time temperature and countdown; (2) the 32-bit microprogrammed control unit (MCU) automatically integrates the reception, processing and execution of electronic signals in a programmable manner, and can be remotely turned on or off over the mobile terminal via a relay transmitter; (3) a lithium battery; (4) an analog digital converter (ADC) that converts an input voltage signal to an output digital signal; (5) a metal-oxide-semiconductor field-effect transistor (MOS FET) switch and a low dropout (LDO) linear voltage regulator to ensure the low voltage direct current (LVDC) over both input and output ports on the motherboard; (6) a negative temperature coefficient (NTC) film temperature sensor, which converts the real-time temperature into MCU-

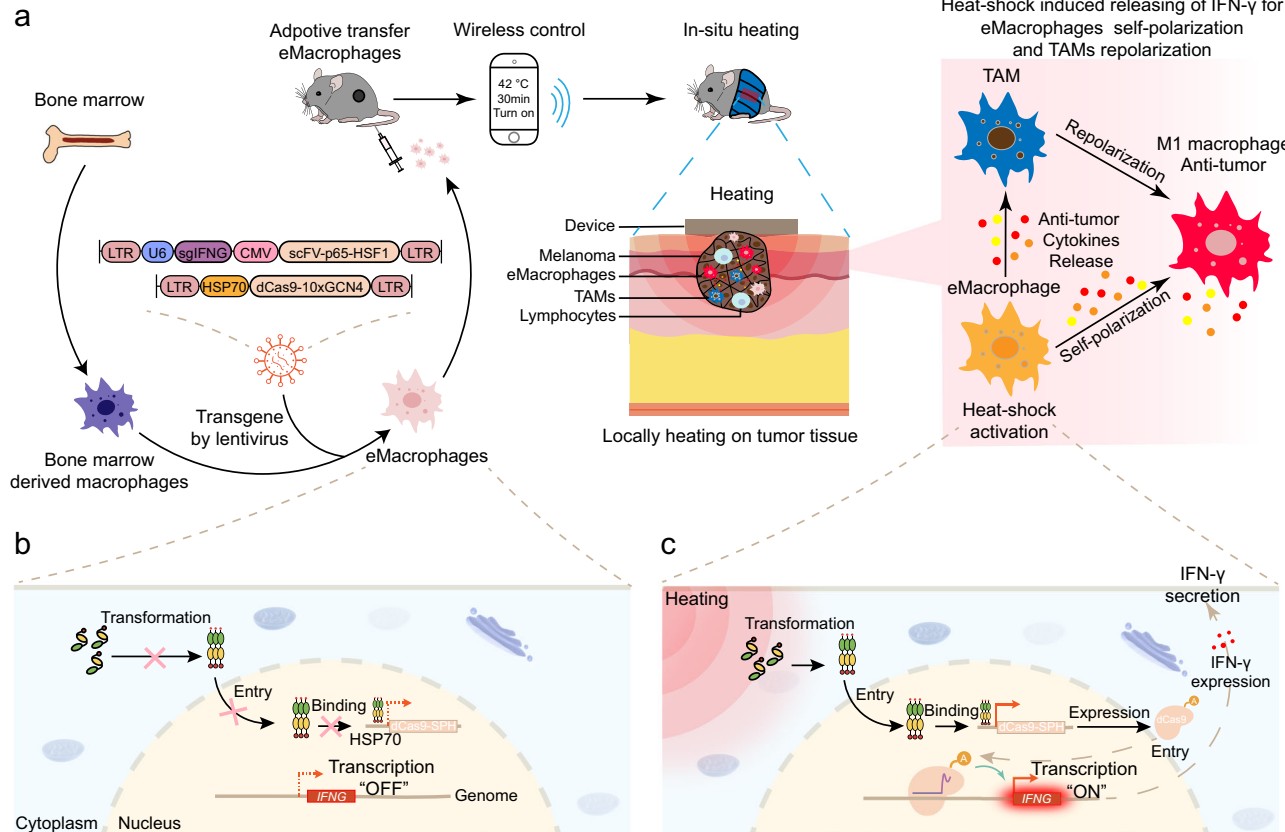

**Fig. 1 | Schematic illustration of wireless remote control of macrophage polarization by an intelligent warming device (iWarm) for melanoma immunotherapy. a** Engineering of BMDMs and polarization of macrophages by smartphone-controlled iWarm for tumor immunotherapy. Mechanism of inducible heat shock (HS) regulation of dCas9-based transcriptional activation of *Ifng* and IFN-γ secretion before (**b**) and after heating (**c**). Hyperthermia in the intracellular

microenvironment induces the transformation of the heat shock factor from inactive monomers to active trimers that are able to translocate into the nucleus. Subsequently, the binding between the intranuclear trimers and the heat shock element of the HSP70 promoter results in the transcription of dCas9 system. The assembled CRISPR/dCas9 gene-regulation system then activates IFN-γ expression.

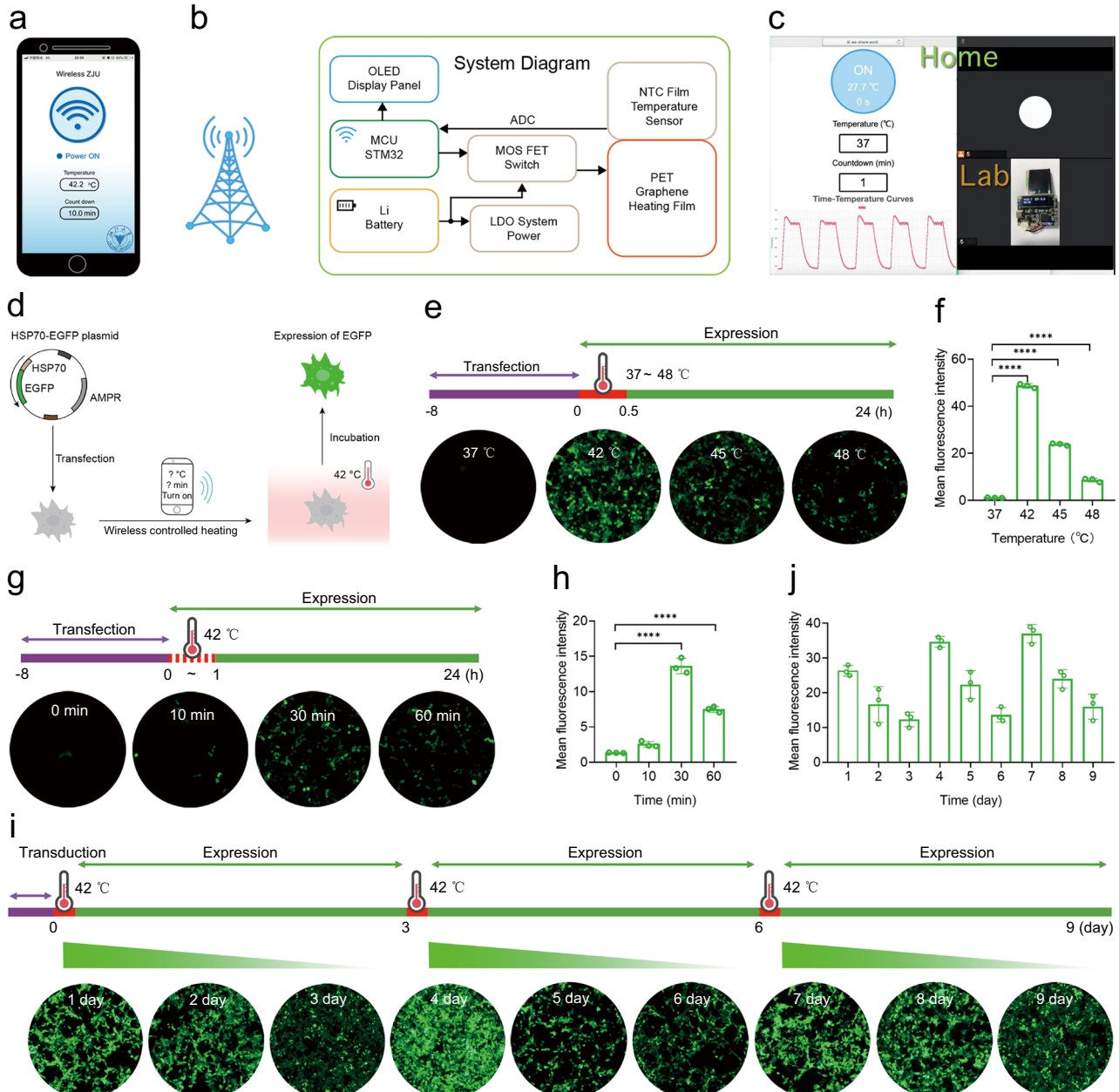

**Fig. 2 | Characterization of the wireless-controlled heat-inducible gene expression. a** The interface of application (APP) for the remote control of iWarm. **b** Detailed electric circuit diagram of the iWarm. **c** The remote control of iWarm from home via internet and the real-time temperature cycle curve of iWarm after heating and cooling of five cycles in lab. **d** Illustration of wireless control of EGFP expression in macrophages. **e** RAW264.7 cells were transfected with HSP70-EGFP plasmids and cultured for 8 h before being heated at different temperature for 30 min controlled by a smartphone. This experiment was repeated three times independently with similar results. The positive EGFP cells were evaluated 24 h after the heat shock and quantitative analysis of EGFP fluorescence by ImageJ (**f**). *n* = 3 biologically independent samples. EGFP expression after wireless-controlled heating at 42 °C for different duration (**g**), and quantitative analysis of EGFP fluorescence 24 h after the heating (**h**). This experiment was repeated three times independently with similar results. **i** Illumination time-dependent HS-mediated ON-OFF kinetics of transgene expression. By lentiviral transduction, EGFP expression was monitored every day after the heat shock at 42 °C for 30 min, which was carried at day 0, day 3, and day 6, respectively. This experiment was repeated three times independently with similar results. The EGFP-positive cells were further quantified by ImageJ (**j**). *n* = 3 biologically independent samples. Data are presented as mean ± SD and statistical significance was calculated via one-way analysis of variance (ANOVA) with Dunnett's multiple comparison tests in (**f**) and (**h**). *P < 0.05; **P < 0.01; ***P < 0.001; ****P < 0.0001. Source data are provided as a Source Data file.

compatible information using an analog-to-digital converter (ADC); (7) a piece of polyethylene terephthalate (PET)/graphene heating film, which converts electric energy into thermal energy in the form of far-infrared radiation (Fig. 2b). As a thermally conductive material, graphene exhibits remarkable electronic and thermal properties, exhibiting a thermal conductivity of ~4000 W·m$^{-1}$·K$^{-1}$[22]. Additionally, graphene exhibits high figure of merits, making it easier to convert

electrical current to heat energy. Moreover, the flexible mechanical properties are ideal for developing graphene as wearable device[22]. Once turning on iWarm from the APP at one location (e.g., home), we found that it quickly heated up to the preset temperature at the other location (e.g., lab), and could be cooled down to room temperature by switching off APP (Supplementary Movie 1 and 2). The repeated heating and cooling of five cycles resulted in a similar temperature

fluctuation, which can be also simultaneously monitored through the APP (Fig. 2c and Supplementary Movie 1). Supplementary Fig. 1a shows the prototype of iWarm that is readily wrapped onto the mouse body through a tailored overcoat. The locoregional temperature of the mouse back quickly increased to 42 °C after turning on iWarm (Supplementary Fig. 1b). To explore the transmission of heat energy through the tissue, the direct heating over a piece of chicken breast tissue (with the thickness of 1 cm) and the temperatures of both front and back sides of the chicken breast tissue were explored (Supplementary Fig. 1c). As monitored by the infrared thermal camera, the temperature of contact surface (front side) reached to 41.9 °C immediately, whereas the non-contact surface (back side) also rose to 41.7 °C in 5 min (Supplementary Fig. 1d).

To explore whether the heat energy generated by iWarm can activate the engineered macrophage, we first transfected RAW264.7 cells with the plasmid encoding EGFP (enhanced green fluorescence protein) reporter driven by the HSP70 promoter using Lipofectamine (Fig. 2d). Then, the activation of EGFP expression was carried out by means of wireless-controlled heating at different temperature or duration. As shown in Fig. 2e, f, the engineered RAW264.7 cells merely showed any green fluorescence at room temperature but become highly fluorescent when the temperature of culture medium increased to 42 °C. Nevertheless, the fluorescence intensity rapidly dropped when the temperature was above 42 °C. At 42 °C, the optimal duration of heat activation was about 30 min (Fig. 2g, h). This was partially supported by our previous results and that of others where the transfection of CRISPR/Cas9 plasmid DNA driven by a heat shock promoter was successfully carried out with the temperature elevation in different cell lines[23–25]. It should be noted that the EGFP expression of engineered RAW264.7 cells regulated by the heat shock was reversible, and the multiple circles of heat shock well re-activated the attenuated EGFP expression, indicating that the gene circuit is sensitive and reversible with ON/OFF kinetics (Fig. 2i, j). Furthermore, we analyzed the ON/OFF ratio and the leaky effect of the gene circuit. When the iWarm was turned on, EGFP expression significantly increased by about 41.7-fold (Supplementary Fig. 2a), and the background leakiness of the gene circuit was only about 2.4% (Supplementary Fig. 2b). In addition, the viability of B16F10, BMDM, and RAW264.7 cells were evaluated after the heat shock (HS) at different temperature or duration (Supplementary Fig. 3a–f). The results showed that the temperature at 42 °C lasted for 30 min merely affected the cell viability of RAW264.7 or BMDM or B16F10 cells. Furthermore, we found that the temperature change could not alter the phenotype of BMDM (Supplementary Fig. 4a, b). Then, we investigated the effect of different temperature on M2 macrophage (IL-4-treated BMDM) phenotype. Similarly, the temperature at 42 °C for 30 min did not alter the phenotype of M2 macrophage in vitro (Supplementary Fig. 4c, d). These results suggested that iWarm-controlled gene expression of macrophages was reversible and could be regulated temporally.

## HS polarizes eMac into a M1 phenotype in vitro

Targeting TAMs represents a viable approach for treating a wide range of cancers and have been extensively investigated in recent years. It is well documented that macrophages are plastic, and their phenotypes and functions are tightly regulated by their surrounding microenvironments[8]. With bioinformatic analysis (The Gene Expression Profiling Interactive Analysis 2 (GEPIA2), http://gepia2.cancer-pku.cn), we found that higher *CD86* or *IFNG* gene expression was associated with longer overall survival of melanoma patients (Supplementary Fig. 5a, b). Moreover, there is a strongly positive correlation between the expression of *IFNG* and *CD86* in melanoma tissues (Supplementary Fig. 5c). CD86 is known as a marker of M1 macrophages which is associated with antitumor effects, whereas IFN-γ is a cytokine encoded by *IFNG* that can stimulate macrophages to polarize into a M1 phenotype[26]. Thus, the functional re-polarization of TAMs into a M1 phenotype in response to

IFN-γ in TME is a promising antitumor strategy. Therefore, we constructed engineered bone marrow-derived macrophages (eBMDM) in which deactivated Cas9 (dCas9)-mediated transcriptional activation system controlled by a heat shock promoter was installed to activate *Ifng* gene expression by wireless-controlled iWarm (Fig. 3a). To explore the performance of iWarm on the polarization of macrophages, the phenotype and function of eMac after iWarm-mediated heat activation were investigated. First, we found that both eBMDM and engineered Raw264.7 (eRaw264.7) were able to secrete IFN-γ continuously for up to 72 h after heating (Fig. 3b, c and Supplementary Fig. 6a). The increment of IFN-γ was further detected after the second heat shock (Fig. 3c). Then, we detected the polarization of M1 macrophages by flow cytometry and RT-qPCR. The results showed that the heat shock could remarkably polarize eBMDM into M1 phenotype, with the considerable increase of CD86+ macrophages, upregulation of M1 markers (*Ifng, Cd86, Il6, Ccl2,* and *Tnf*), and downregulation of M2 markers (*Cd206, Il10, Arg1,* and *Fizz1*) (Fig. 3d–g). Similarly, eRaw264.7 exhibited similar profiles in terms of CD86+ macrophages, M1 and M2 markers (Supplementary Fig. 6b–e). We further explored the performance of iWarm on the effector function of macrophages. Firstly, we examined whether the cytokines secreted by macrophages could inhibit the growth of tumor cells. As shown in Fig. 3h, whereas eBMDM were cultured in the upper chamber of Transwell system, B16F10 cells were separately cultured in the lower chamber. After heat shock at 42 °C for 30 min, both cells are continuously cultured for 24 h before the cell viability of B16F10 cells was examined. As expected, the number of viable B16F10 cells significantly decreased, suggesting the cytokine released from eBMDM or eRaw264.7 after heat shock contributes to the inhibition of growth of tumor cells (Fig. 3i and Supplementary Fig. 6f). We then investigated the phenotype of eBMDM in Transwell system by flow cytometry (Supplementary Fig. 7). As compared with BMDM cultured alone (NC group), control group, HS group and eBMDM group (without heat shock) were slightly polarized into M1 phenotype, and engineered BMDM after heat shock could be significantly polarized into M1 (Supplementary Fig. 7). To investigate the phagocytic capability of eMac, we labeled eBMDM or eRaw264.7 and B16F10 cells with DiI (1,1′-dioctadecyl-3,3,3′,3′-tetramethylindocarbocyanine perchlorate, red) and CFSE (carboxyfluorescein diacetate, succinimidyl ester, green) staining dye, respectively. The co-culture of eBMDM or eRAW264.7 with B16F10 cells suggested the greater phagocytic capability of these engineered macrophages after the heat shock, as compared with those without heat shock (Fig. 3j–l and Supplementary Fig. 6g, h). To verify the phagocytosis of tumor cells by macrophages, we detected the capability of BMDM for phagocytosis in a co-culture system, in which CM-DiI-labeled BMDM were cultured with CD47-knockdown B16F10 cells. The result indicated that the knockdown of CD47 in B16F10 cells by siRNA significantly increased the phagocytic capability of BMDM (Supplementary Fig. 8), suggesting that the phagocytic pathway plays an important role by which macrophages exert antitumor effects. The above results were further confirmed by confocal laser scanning microscopy imaging results, where the green fluorescence was highly embraced by red fluorescence in many cases (white arrows) (Fig. 3m, Supplementary Figs. 6i and 9). Finally, we tested the tumor tropism of eBMDM or eRAW264.7 by Transwell assay. Both types of macrophages could migrate through the semipermeable membrane to contact B16F10 cells in a similar way as native macrophage (without engineering), suggesting the tropism of eBMDM or eRAW264.7 toward tumor cells was independent of genetic engineering (Fig. 3n–p and Supplementary Fig. 6j–k).

## Inhibition of tumor growth and lung metastasis by thermo-responsive eMac

To investigate the distribution of eBMDM or eRAW264.7 in vivo after systemic administration, DiI-labeled eBMDM or eRAW264.7 and luciferase-expressing eBMDM or eRAW264.7 were adoptively

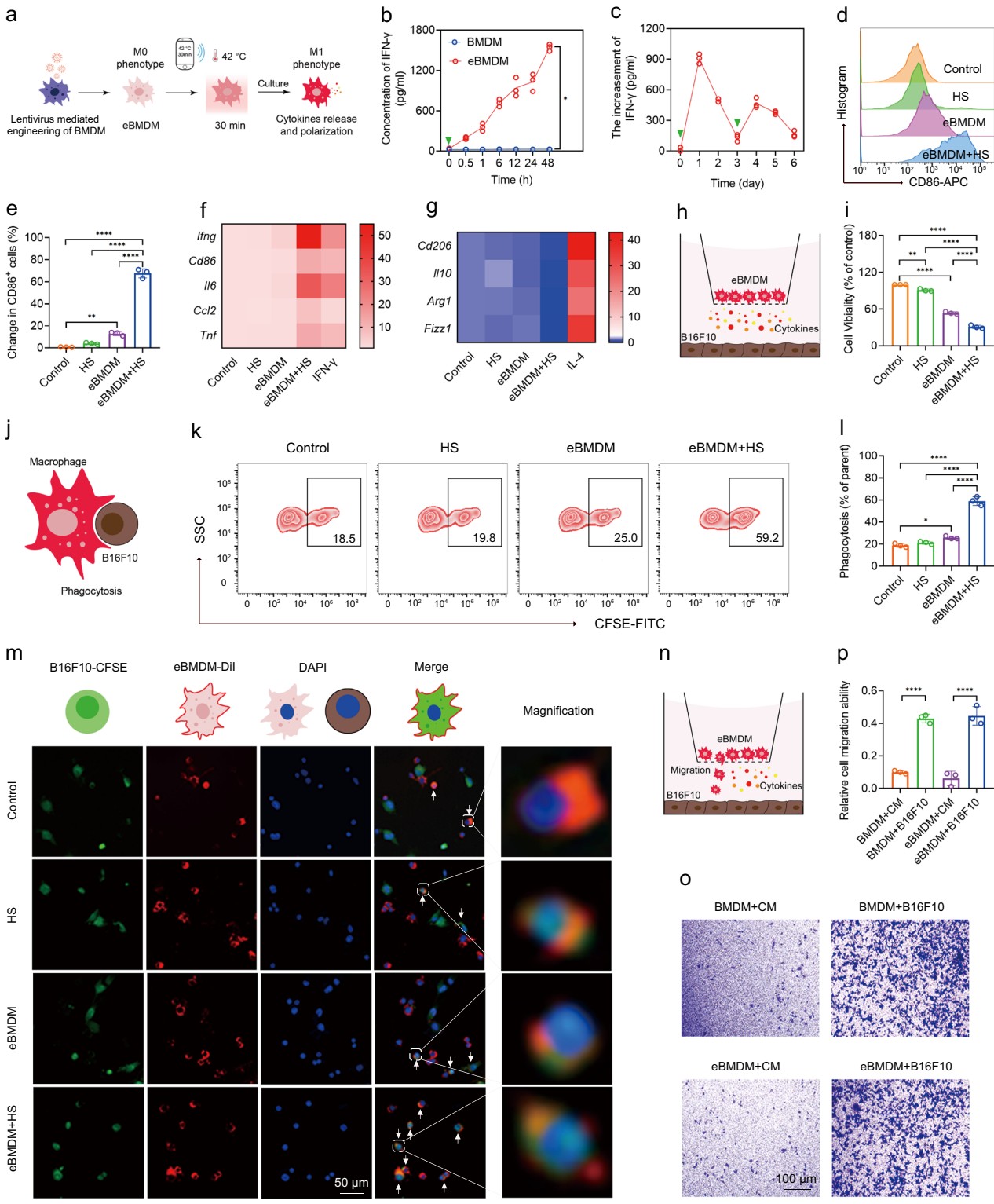

transferred and the fluorescence intensity was evaluated at different time points. In general, the adoptively transferred eBMDM or eRAW264.7 primarily distributed in the liver, lung and tumor, and the fluorescence was clearly observable in the tumor even after 5 days, suggesting the strong tropism of eBMDM or eRAW264.7 toward tumors in vivo (Fig. 4a, Supplementary Fig. 10a, Supplementary Figs. 11a, b and 12a). When the tumor was heated by wireless-enabled iWarm, the strong luminescence was observed locoregionally after the systemic administration of luciferase-expressing eBMDM or eRAW264.7 driven by the heat shock promoter (Fig. 4b,

Supplementary Figs. 10b and 12b), suggesting the eBMDM or eRAW264.7 were highly sensitive to locoregional hyperthermia. To investigate the heat shock-responsiveness efficiency of HSP70 promotor in vivo, after the systemic administration of luciferase-expressing eBMDM or eRAW264.7 driven by the heat shock promoter, the luminescence intensity in the tumor was evaluated once a day the after hyperthermia activation for different times. The results showed that the luminescence signal enhanced with the increased heating times (Fig. 4c and Supplementary Fig. 10c). We also investigated the viability duration of eBMDM or eRAW264.7 in vivo, and

**Fig. 3 | Heat shock (HS) polarizes engineered BMDMs (eBMDM) into a M1 phenotype in vitro. a** Illustration of engineering process of BMDM and the polarization of eBMDM by wireless-controlled secretion of IFN-γ. **b** The increase of IFN-γ by eBMDM after HS at 42 °C for 30 min. **c** The increment of IFN-γ by eBMDM after HS at 42 °C for 30 min at day 1 and day 3, respectively. The green arrow in (**b**) and (**c**) refers to the time point of HS. **d** Flow cytometry analysis and (**e**) quantitative analysis of CD86+ macrophages in eBMDM after the indicated treatment. RT-qPCR analysis of M1 macrophages markers (**f**) and M2 macrophages markers (**g**) after the indicated treatment. IFN-γ treated BMDMs and IL-4-treated BMDMs were used as positive controls. **h** Illustration of proinflammatory cytokines-mediated inhibition of tumor proliferation by using Transwell system. **i** Proinflammatory cytokines-mediated inhibition of tumor proliferation was determined by CCK8 assay with the indicated treatment. **j** Illustration of the phagocytosis of B16F10 cells by polarized M1 eBMDM. **k, l** Flow cytometry and quantitative analysis of the phagocytosis of

BMDMs after the specified treatment. **m** Fluorescence images of the phagocytosis by BMDMs. B16F10 cells were labeled with CFSE (green), the cell membrane of eBMDM was labeled with DiI (red), and the nuclei of both cells were stained with DAPI (blue). The white arrows point to the B16F10 cell phagocytized by macrophages. **n** Illustration of the chemotaxis of eBMDM by Transwell assay. **o** The tumor-targeting tropism was evaluated by the number of BMDMs that migrates through the semipermeable membrane. **p** Quantitative analysis of cell migration ability in (**o**) after the specified treatment. CM stands for culture medium. Data are presented as mean ± SD, $n = 3$ biologically independent samples in (**b–o**). Statistical significance was calculated via two-tailed paired *t*-test in (**b**) and one-way ANOVA with a Sidak's multiple comparisons in (**e**) and one-way ANOVA with a Tukey's multiple comparisons test in (**i**), (**l**) and (**p**). *$P < 0.05$; **$P < 0.01$; ***$P < 0.001$; ****$P < 0.0001$. Source data are provided as a Source Data file.

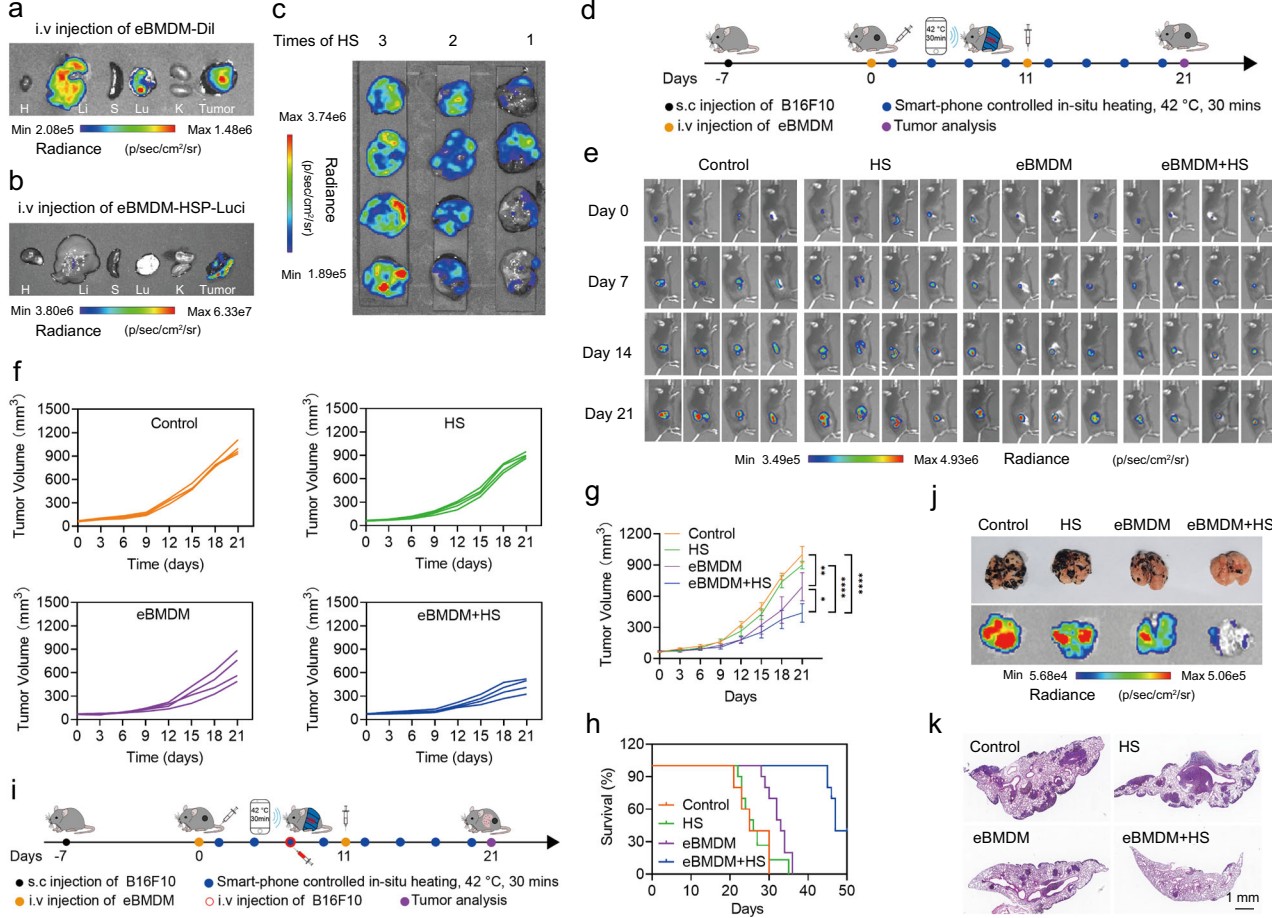

**Fig. 4 | eBMDM-mediated treatment in vivo controlled by iWarm.**
**a** Fluorescence image of the eBMDM distribution in major organs and the tumor tissue after the intravenous (*i.v.*) injection of DiI-labeled eBMDM for 24 h.
**b** Bioluminescence image of major organs and tumor tissue after the *i.v.* injection of HSP-luciferase-expressing eBMDM for 24 h, followed by locoregional hyperthermia in tumor tissue. **c** Bioluminescence image of tumor tissue after *i.v.* injection of HSP-luciferase-expressing eBMDM, followed by locoregional hyperthermia in tumor tissue for different times. **d** Illustration of B16F10 tumor therapy in vivo with eBMDM via remote control of locoregional hyperthermia. **e** In vivo bioluminescence images of mice after the specified treatment at day 0, day 7, day 14, and day 21, where B16F10 cells were tagged with luciferase. **f** Individual and **g** average tumor

growth curves after the specified treatment. **h** Survival curves of mice after the specified treatment. **i** Illustration of wireless-controlled eBMDM for treating B16F10 tumor metastasis in vivo. **j** Bioluminescence image of lung metastatic nodules of the B16F10 tumors after the treatment. **k** H&E staining of lung metastatic nodules of B16F10 tumor after the treatment. Data are presented as mean ± SD, $n = 3$ biologically independent mice or samples in (**a**), (**b**), (**j**), (**k**); $n = 4$ biologically independent mice or samples in (**c–g**) and $n = 10$ biologically independent mice in (**h**). Statistical significance was calculated via one-way ANOVA with a Tukey's multiple comparisons test in (**g**). *$P < 0.05$; **$P < 0.01$; ***$P < 0.001$; ****$P < 0.0001$. Source data are provided as a Source Data file.

found that eBMDM could survive for about 7 days in the tumor tissue (Supplementary Fig. 11a) and eRAW264.7 could survive for about 9 days in the tumor tissue (Supplementary Fig. 11b). In addition, we compared the efficacy of IFN-γ and eBMDM followed by iWarm-enabled locoregional hyperthermia in the treatment of melanoma

(Supplementary Fig. 13a). The systemic administration of IFN-γ could neither inhibit the tumor growth, nor increase the IFN-γ level in the tumor tissues. In contrast, eBMDM followed by iWarm-enabled locoregional hyperthermia could significantly inhibit the tumor growth with the observable increase of IFN-γ level in the tumor tissue

(Supplementary Fig. 13b–d). The treatment by administrating eBMDM with two shots (on day 0 and day 11, respectively) was more effective than one-shot treatment against melanoma (Supplementary Fig. 13b–d). Our results also showed that the systemic injection of eBMDM or eRAW264.7 followed by iWarm-enabled locoregional hyperthermia could significantly inhibit the tumor growth (Fig. 4d–g and Supplementary Fig. 12c–f) and greatly extended the survival time (Fig. 4h and Supplementary Fig. 12g), which was more superior to other groups. Using B16F10 tumor model with lung metastasis (Fig. 4i), we noticed that eBMDM+HS or eRAW264.7+HS treatment group substantially decreased the metastatic tumor nodules in the pulmonary tissue, as reflected by the luciferase intensity (Fig. 4j and Supplementary Fig. 12h) and H&E staining of lung tissues (Fig. 4k and Supplementary Fig. 12i), with less metastatic nodules observed in the lung. The results validate the effectiveness of eMac for the treatment of melanoma under the iWarm-enabled locoregional hyperthermia. The mouse skin treated with iWarm up to a month suggested no tissue injury or damage, as indicated by H&E staining (Supplementary Fig. 14). After eBMDM+iWarm or eRAW264.7+iWarm treatment, there is no toxicity to the major organs, as further demonstrated by H&E staining (Supplementary Fig. 15), and hematological evaluation suggested that the treatment neither caused any damage to the liver and kidney nor resulted in any inflammation (Supplementary Fig. 16), suggesting the safety of such a treatment modality. These results highlighted the translational potential of engineered macrophages under the heat control for precision cell therapy.

## Self-polarization of eMac and re-polarization of TAMs by iWarm
To investigate the infiltration ratio of eBMDM in tumor tissue in vivo, CM-DiI-labeled eBMDM were adoptively transferred and the infiltration of eBMDM in melanoma tissue was evaluated by flow cytometry. First, we found that adoptive eBMDM accounted for ~3.8% of the total macrophages (TAMs) (Supplementary Fig. 17a, b). As compared with other groups, eBMDM+HS or eRAW264.7+HS treatment could substantially induce the M1 phenotype of TAMs, with the observable upregulation of M1 markers (*Ifng*, *Cd86*, *Il6*, *Ccl2* and *Tnf*) (Fig. 5a and Supplementary Fig. 18a) and downregulation of M2 markers (*Cd206*, *Il10*, *Arg1*, *Fizz1*) (Fig. 5b and Supplementary Fig. 18b) as well as increased M1/M2 ratio (Fig. 5c, d and Supplementary Fig. 18c, d). Flow cytometry analysis showed that eBMDM+HS treatment could substantially reduce the number of M2-like TAMs as compared with other groups (Supplementary Fig. 17c), and the percentage of adoptive eBMDM in M2-like TAMs increased substantially from 14.6% (eBMDM group) to 65% (eBMDM+HS group) due to the decreased M2-like TAMs (Supplementary Fig. 17d). Then, we accessed the ratio of M1-like TAMs and the ratio of M1 eBMDM in M1-like TAMs population by flow cytometry. The result showed that eBMDM+HS treatment could substantially induce the M1 phenotype of TAMs as compared with other groups (Supplementary Fig. 17e), and the ratio of adoptive eBMDM to M1-like TAMs accounted for ~16% in eBMDM+HS group (Supplementary Fig. 17f). Also, we found that M1 eBMDM accounted for ~4.9% of the total population of M1-like TAMs (Supplementary Fig. 17g, h), and the substantial increase in the ratio of CD8$^+$ T cells was also detected in eBMDM+HS or eRAW264.7+HS group (Fig. 5e–g and Supplementary Fig. 18e–g). In addition, whereas the antitumor cytokines, like IFN-γ and TNF-α, in melanoma tissues increased, both IL-4 and IL-10 (pro-tumor cytokines) levels decreased at the end of the indicated treatment (Fig. 5h, i and Supplementary Fig. 18h, i). After eBMDM+HS treatment, though IFN-γ level significantly increased in tumor tissue compared with other groups, the level in serum remains unchanged (Supplementary Fig. 19). We also detected the IFN-γ levels in tumor tissues after the treatment at different time point (Supplementary Fig. 20a). After eRAW264.7+HS treatment, we observed a significant increase of IFN-γ after HS in two days and the IFN-γ level decreased to baseline level without HS in

4 days (Supplementary Fig. 20b). Immunofluorescence that is intended to access the infiltration of M1-like TAMs into the melanoma tissue, indicated that eBMDM+HS not only promoted the infiltration of M1 eBMDM (orange) into the tumor tissue, but also induced in-situ macrophages (TAMs) into M1 phenotype (green) as a result of re-polarization (Fig. 5j, Supplementary Figs. 18j and 21a–c). It should be noted that those macrophages from adoptive transfer also polarized into M1 phenotype due to self-polarization. In the meantime, M2 marker of the macrophages, CD206, also considerably decreased after eBMDM+HS treatment (Fig. 5k, Supplementary Figs. 18k and 21d). These results collectively indicated the success of self-polarization of eBMDM or eRAW264.7 as well as re-polarization of TAMs into M1 phenotype under the iWarm-enabled locoregional hyperthermia.

## Macrophage depletion attenuates antitumor activity of adoptive eMac therapy
To further confirm the re-polarization of TAMs by eBMDM, we evaluated the treatment effectiveness over the macrophage-deficient melanoma models in which macrophages were depleted by clodronate liposomes (Clo) in vivo (Fig. 6a). First, we found that Clo treatment inhibited tumor growth, possibly due to the decreased number of TAMs. In addition, Clo treatment also attenuated the antitumor activity of eBMDM with iWarm (Fig. 6b–d), and the survival of tumor-bearing mice was shortened after eMac+HS treatment in the presence of Clo, as compared with that by eBMDM+HS treatment in the absence of Clo (Fig. 6e). Next, we accessed the infiltration of macrophages in melanoma tissue by flow cytometry (Fig. 6f, g). The Clo treatment significantly depleted macrophages compared with the control group, while the ratio of macrophages in the tumor increased following adoptive eBMDM therapy compared with Clo treatment group. Furthermore, the immunofluorescence was also performed to assess the infiltration of macrophages in the melanoma tissue. In agreement with the results of flow cytometry, Clo treatment could significantly deplete macrophages in tumor tissue, whereas the adoptive transfer of eBMDM could supplement the macrophages and increase the amounts of macrophages in the tumor (Fig. 6h). In addition, the infiltration ratio of CD4$^+$ and CD8$^+$ T cells in the melanoma tissue was also detected by flow cytometry. The Clo treatment alone could slightly increase the infiltration ratio of CD4$^+$ and CD8$^+$ T cells in the tumor tissue, but CD4$^+$ and CD8$^+$ T cells decreased in the eBMDM+HS group following Clo treatment as compared with the same group without Clo treatment (Fig. 6i–k). Finally, we observed IFN-γ level decreased in eBMDM+HS+Clo group compared with eBMDM+HS group (Fig. 6l). These results demonstrate that the re-polarization of TAMs by eMac under the iWarm-enabled locoregional hyperthermia plays an important role in antitumor therapy.

## IFN-γ neutralization attenuates antitumor activity of adoptive eBMDM therapy mediated by wireless-controlled iWarm
To prove that heat-inducible IFN-γ expression contributes to the efficacy of the treatment, we used the anti-mouse IFN-γ antibody to neutralize IFN-γ in vivo (Fig. 7a). First, we found that anti-IFN-γ treatment attenuated the antitumor activity of eBMDMs+HS treatment (Fig. 7b–d), and IFN-γ level decreased in the melanoma tissues after anti-IFN-γ treatment, as compared with control group treated with isotype control antibody (Fig. 7e). In the meantime, IFN-γ level also decreased in eBMDM+HS+anti-IFN-γ group as compared with eBMDM+HS group (Fig. 7e). Then, we accessed the infiltration of M1-like and M2-like TAMs in melanoma tissue by flow cytometry. Of note, the ratio of M1-like TAMs significantly decreased while the ratio of M2-like TAMs macrophages significantly increased after eBMDM+HS treatment in the presence of anti-IFN-γ, as compared with the treatment by eBMDM+HS (Fig. 7f–i). Furthermore, the ratio of CD8$^+$ T cells significantly decreased in the eBMDM+HS+anti-IFN-γ group, as compared with eBMDM+HS group (Fig. 7j, k). Collectively, these results demonstrate

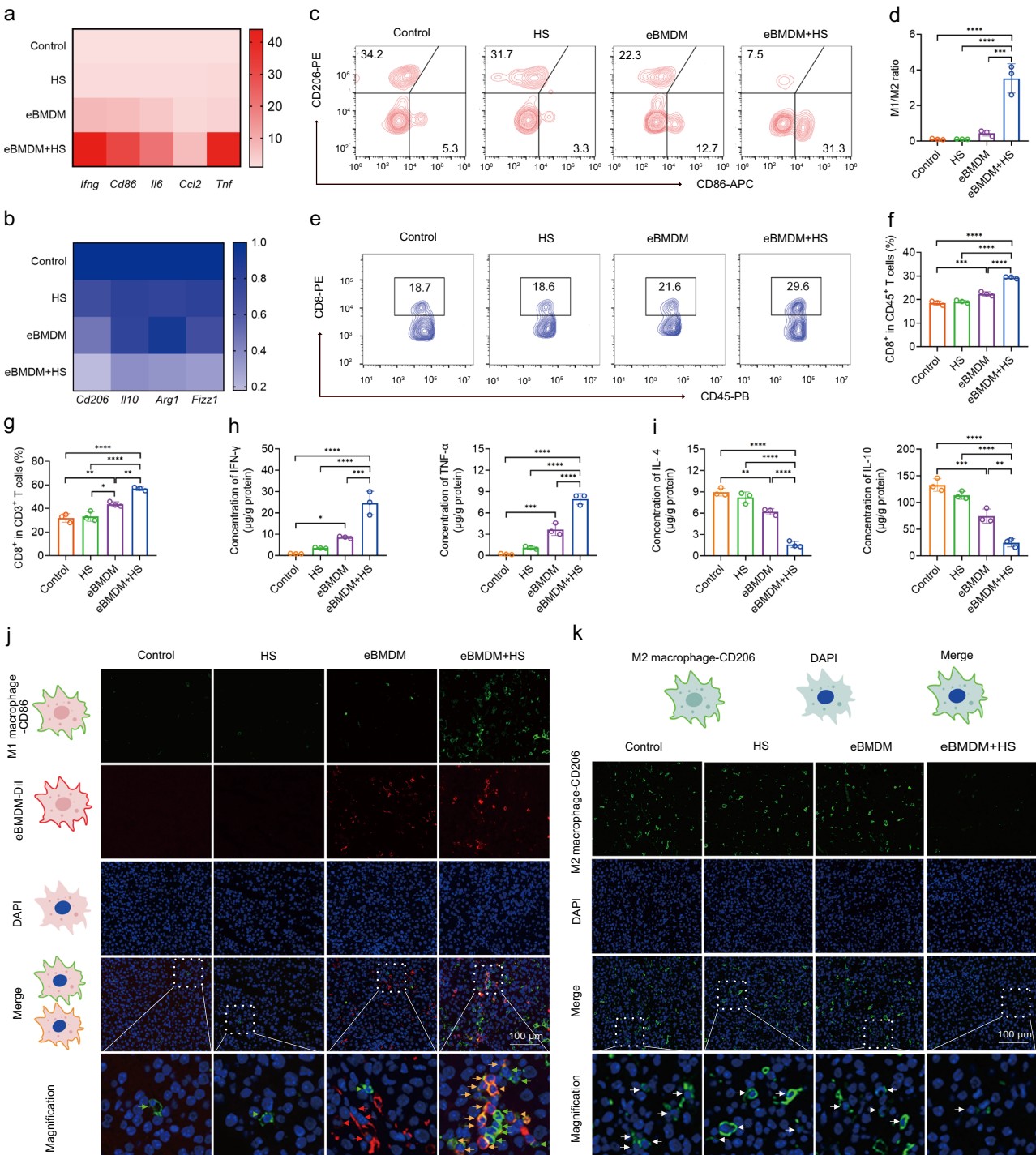

**Fig. 5 | Wireless-controlled iWarm mediated the polarization of eBMDM and re-polarization of TAMs into a M1 phenotype, which trigger robust antitumor immunity in vivo.** RT-qPCR analysis of M1 (**a**) and M2 (**b**) macrophages markers in tumor tissues after the indicated treatment. **c** Flow cytometry analysis of the polarization of macrophages in tumor tissues after the indicated treatment. CD86+ is the marker of M1 macrophages, while CD206+ is the marker of M2 macrophages. **d** Quantitative analysis of the M1/M2 ratio of macrophages in (**c**). **e** Flow cytometry analysis the CD8+ T cells in tumor tissues after eBMDM treatment with or without heat shock. **f**, **g** Quantitative analysis of the ratio of CD8+ T cells in CD45+ cells and CD3+ cells. **h** IFN-γ and TNF-α levels in tumor tissues collected from mice after the indicated treatment. **i** IL-4 and IL-10 levels in tumor tissues collected from mice after the indicated treatment. **j** Multiplex IHC images of M1 macrophage and eBMDM infiltration in tumor tissues. In merged figures, DiI-positive and CD86-positive cells (orange) represent M1 eBMDM, CD86-positive only cells (green) represent re-polarized TAMs. **k** IHC images of M2 macrophage infiltration in tumor tissues. Data are presented as mean ± SD, *n* = 3 biologically independent samples in (**a**–**k**). Statistical significance was calculated via one-way ANOVA with a Sidak's multiple comparisons in (**d**) and one-way ANOVA with a Tukey's multiple comparison tests in (**f**–**i**). \**P* < 0.05; \*\**P* < 0.01; \*\*\**P* < 0.001; \*\*\*\**P* < 0.0001. Source data are provided as a Source Data file.

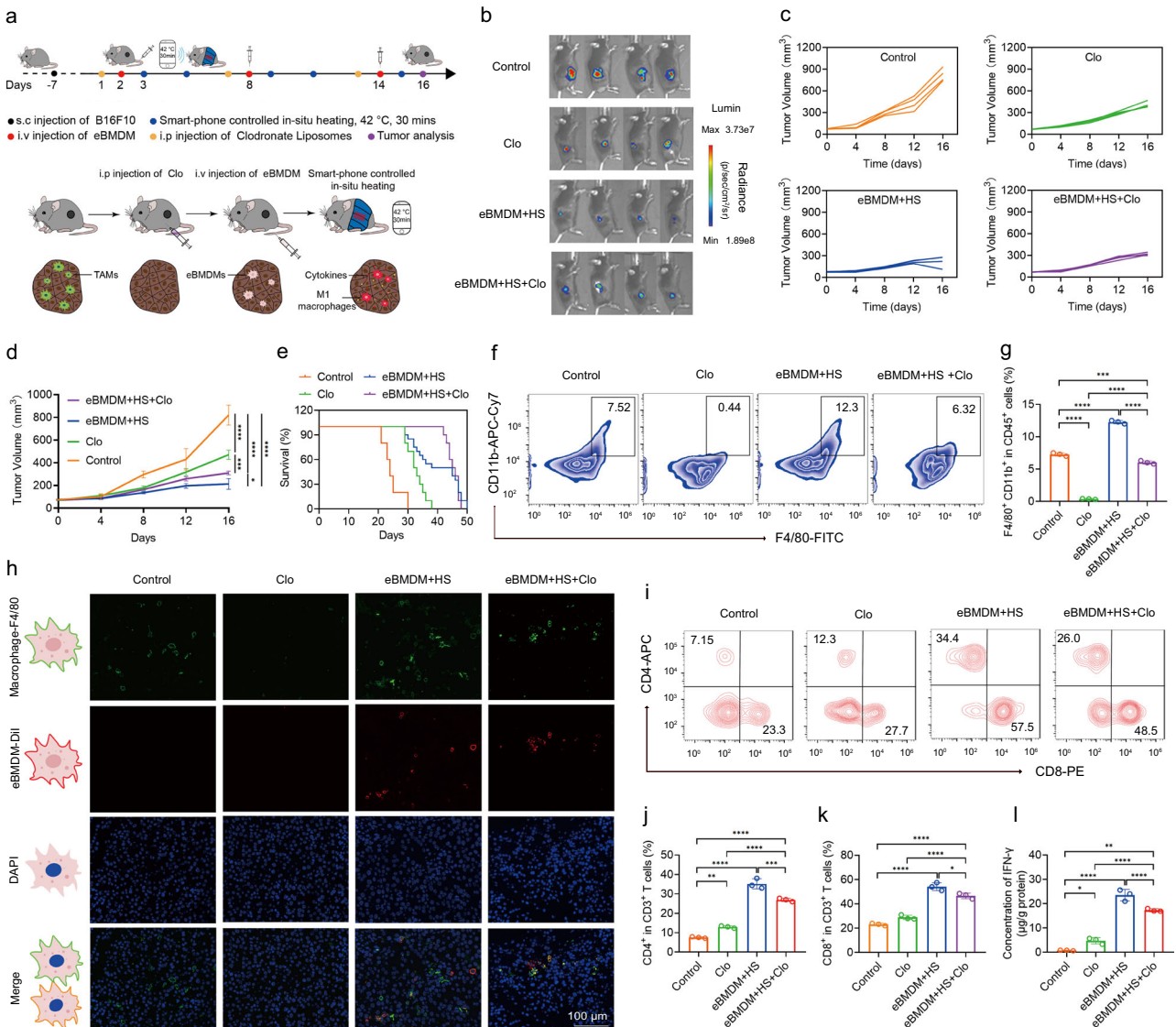

**Fig. 6 | Macrophage depletion attenuates antitumor activity of adoptive eBMDM therapy mediated by wireless-controlled iWarm. a** Illustration of macrophage depletion and adoptive eBMDM therapy with wireless iWarm in vivo. **b** In vivo bioluminescence images of mice after the specified treatment at day 16. **c, d** The inhibition of tumor growth of tumor-bearing mice after the indicated treatment. **e** The survival curve of tumor-bearing mice after the indicated treatment. **f** Flow cytometry analysis the macrophages infiltration in tumor tissues after the indicated treatment. CD11b+ F4/80+ is the marker of macrophages. **g** The quantitative analysis of F4/80+ CD86+ macrophages in tumor tissues after the indicated treatment by flow cytometry. **h** Multiplex IHC images of macrophage and

eBMDM infiltration in tumor tissues. **i** Flow cytometry analysis the CD4+ T cells and CD8+ T cells in tumor tissues after the indicated treatment. The quantitative analysis of the ratio of CD4+ T cells (**j**) and CD8+ T cells (**k**) in tumor tissues after the indicated treatment by flow cytometry. **l** IFN-γ levels in tumor tissues collected from mice after the indicated treatment. Data are presented as mean ± SD, n = 4 biologically independent mice in (**b**–**d**); n = 10 biologically independent mice in (**e**), and n = 3 biologically independent samples in (**g**–**l**). Statistical significance was calculated via one-way ANOVA with a Tukey's multiple comparison tests in (**d**), (**g**), and (**j**–**l**). *P < 0.05; **P < 0. 01; ***P < 0.001; ****P < 0.0001. Source data are provided as a Source Data file.

that the effectiveness of eBMDMs for the treatment of melanoma under the iWarm-enabled locoregional hyperthermia is largely attributed to the heat-inducible IFN-γ expression.

## Discussion

To date, macrophage-based cell therapy has shown great potentials in treating a wide variety of diseases including cancers, due to their unique effector functions and capability to penetrate into deep tumors as well as their roles in various inflammatory processes[3,27–30]. One of the key challenges associated with macrophage-based cell therapies lies in their high plasticity in response to external stimuli. Although a number of previous efforts, including systemic administration of small molecule inhibitors[31–33], cytokines (GM-CSF, IFN-γ)[15,34], or antibodies[35,36],

etc., have been attempted to polarizing macrophages into antitumoral, proinflammatory M1 phenotype, these biological agents often induce side effects to healthy tissues and elicit undesirable immune responses that severely limit the effective polarization of either adoptively transferred macrophages or TAMs. Therefore, an effective approach for locoregional polarization is of paramount importance to improve the safety and efficacy of macrophage-based cancer immunotherapies. Though several nanoparticle-based approaches have shown promises to deliver biologics into TAMs to induce the polarization, the unavoidable non-specific distribution of these nanoparticles in the mononuclear phagocyte system, such as liver and lung, greatly impairs their efficacy to target TAMs[12]. Indeed, the current study exploits exogenous macrophages as the delivery vector, which bears the

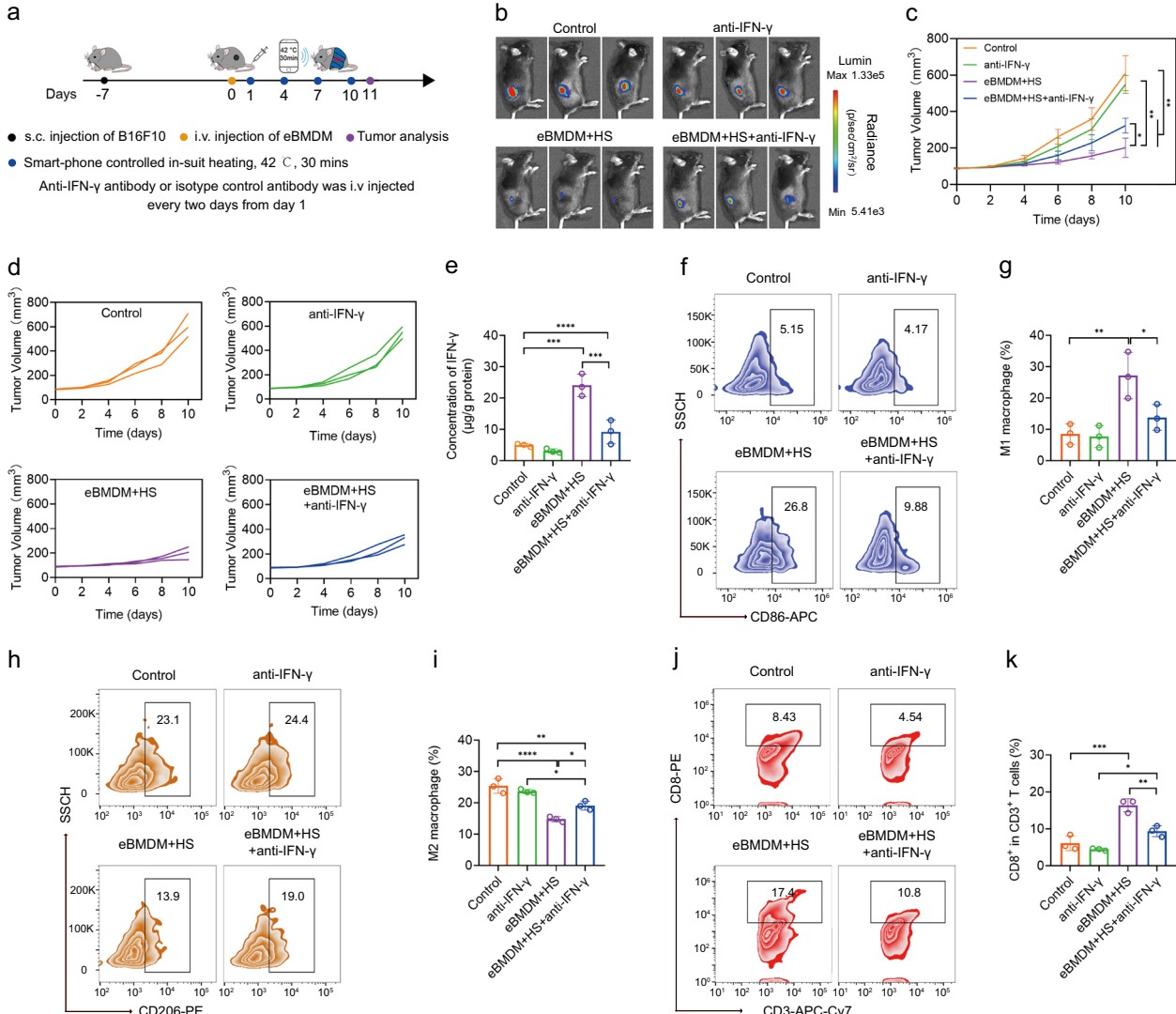

**Fig. 7 | IFN-γ neutralization attenuates antitumor activity of adoptive eBMDM therapy mediated by wireless-controlled iWarm. a** Illustration of IFN-γ neutralization and adoptive eBMDM therapy with wireless iWarm in vivo. **b** In vivo bioluminescence images of mice after the specified treatment at day 11. Average (**c**) and individual (**d**) tumor growth curves after the specified treatment. **e** IFN-γ levels in tumor tissues collected from mice after the indicated treatment. Flow cytometry analysis the M1 polarization (**f**) and the M2 polarization (**h**) of macrophages in tumor tissues after the indicated treatment. CD86⁺ is the marker of M1 macrophages, while CD206⁺ is the marker of M2 macrophages. Quantitative analysis of the M1 (**g**) and M2 (**i**) ratio of macrophages in (**e**) and (**g**). **j** Flow cytometry analysis the CD8⁺ T cells in tumor tissues after the indicated treatment. **k** Quantitative analysis of the ratio of CD8⁺ T cells. Data are presented as mean ± SD, $n = 3$ biologically independent mice or samples in (**a**–**k**). Statistical significance was calculated via two-tailed unpaired $t$-test in (**c**) and one-way ANOVA with a Tukey's multiple comparison tests in (**e**), (**g**), (**i**), and (**k**). *$P < 0.05$; **$P < 0.01$; ***$P < 0.001$; ****$P < 0.0001$. Source data are provided as a Source Data file.

inherent ability to target TME and locoregionally secretes and releases IFN-γ through the remote control to enhance the safety and efficacy of cancer immunotherapy. Such a design combines the targeted cell delivery and precise release of biologics essential for the polarization, thus greatly reducing the systemic toxicity due to 'leaky' effects and non-specific distribution encountered by nanoparticle delivery. As a result, both adoptively transferred, exogenous macrophages and endogenous TAMs can be locoregionally 're-educated' into M1 phenotype for their antitumoral role.

As a new generation of gene-regulation tool, CRISPR/dCas9 system offers a straightforward and precise approach to modulate specific genes simply by designing corresponding single-guide RNA (sgRNA), a major advantage over other traditional regulation approaches of gene transcription. In the current study, we first established an upgraded dCas9 system that was fused with 10 copies of GCN4 (encoding activating transcription factor), which is linked with scFv (single-chain variable fragment)-p65-HSF1, respectively to induce

more potent transcriptional activation[37]. As heat shock promoter allows for the precise control of gene expression in a spatiotemporal manner, the installation of HSP70 promoter to the upgraded dCas9 system ensures the precise regulation of transcriptional activation of *Ifng* in response to mild hyperthermia. Meanwhile, the genetic engineering of macrophages was fulfilled by lentiviral transfection to enable the integration of HSP70-dCas9 system into the genome of macrophage, which allows the macrophage to serve as a cell factory to produce IFN-γ. Of note, the overexpression of IFN-γ from eMac regulated by heat shock was reversible and can be terminated simply through switching off the iWarm to avoid the unwanted side effects, such as cytokine release syndrome and immune effector cell-associated neurotoxicity syndrome that are commonly encountered in other cell therapies, such as CAR-T therapy.

Several transcriptional control systems have been developed for dynamically regulating gene expression in a spatiotemporal way[25,38,39]. Previous have shown that the activity of intratumoral CAR-T cells can

be precisely controlled by either focused ultrasound[40] or photo-thermal approach[41] to trigger the locoregional transgene expression to augment safety and efficacy of cell-based therapies. Despite their effectiveness for the locoregional induction of mild hyperthermia, these devices are generally bulky and have to be operated by trained personnel. In contrast, the wearable iWarm device we have developed is user-friendly, and can be easily operated by any untrained person from their own device (such as computer, mobile phone and smart watches, etc.). Particularly, the wireless control of iWarm provides a remote, online modality for doctors or patients to deliver on-demand medications (Supplementary Fig. 22). This is particularly favorable for those patients, such as elders who are unable to operate by themselves or on time. Thus, one can expect that such a treatment represents a new paradigm of telemedicine, opening the possibility of delivering remote healthcare and medications to patients in their homes, work-places, or even rehabilitation center (Supplementary Fig. 22). In addition, the wearable warming device is fabricated from PET/graphene heating film, which has shown outstanding mechanical properties and thermal conductivity[42]. Such features of PET/graphene film allow the device to be easily customized according to the requirement of patients. Given that the conduction of heat accompanies dis-equilibrium conditions in tissues[43], the temperature elevation by iWarm in deep tissues is limited. Thus, besides melanoma, iWarm is expected to extend to superficial tumor therapies, such as Merkle cell carcinoma and squamous cell carcinoma. Nevertheless, for those deep-seated tumors, such as non-small cell lung carcinoma or pan-creatic ductal adenocarcinoma, the current therapeutic strategy seems to be limited. In this respect, future design should be dedicated to upgrading iWarm by integrating wearable magnetic or ultrasound technologies into iWarm device to sidestep the issue of heat genera-tion at the deep tissue[44–47]. For example, the ultrasound miniaturiza-tion allows for the development of wearable ultrasound device (commercially available as sustained acoustic medicine, sam®) that exhibits temporal heating profiles up to 5-cm tissue depth[48]. Collec-tively, the remote control of engineered macrophages through an intelligent device offers a new therapeutic paradigm for developing personalized, precision medicine.

In summary, we report an engineered macrophage with a heat-inducible genetic switch that can be remotely controlled by a wireless warming device to initiate the locoregional polarization for precision adoptive cell therapy. The genetic engineering is accomplished by assembling an upgraded CRISPR/dCas9 system driven by a heat shock promoter into the macrophages, which allows the ultrasensitive tran-scriptional activation of *Ifng* gene encoded for IFN-γ in response to the mild temperature elevation. Following the adoptive transfer, the intrinsic tumor tropism of eMac contributes to the accumulation in the TME, where the locoregional temperature elevation to ~42 °C by an iWarm direct both self-polarization of transferred eMac and re-polarization of TAMs toward antitumoral phenotype under the locor-egional secretion of IFN-γ from eMac. Such a modality well avoids the rapid clearance, dose-dependent toxicity, and side effects encoun-tered by systemic administration of IFN-γ required for effective polarization. Particularly, iWarm can be customized into wearable device and remotely controlled by a smartphone or an electronic device that can be connected to internet (such as laptop), opening the possibilities to deliver remote and precision medications to patients.

## Methods

### Ethics statement
Our research complies with all relevant ethical regulations. All animal treatments or procedures were carried out in accordance with the guidelines of the Laboratory Animal Welfare and Ethics Committee of Zhejiang University and approved by the Animal Ethics Committee of Sir Run Run Shaw Hospital, School of Medicine, Zhejiang University. The maximal tumor volume permitted in our study is 2000 mm³ as

calculated from caliper tumor measurements. All procedures were performed in accordance with the protocol and none of the tumor volume in our study exceeded 2000 mm³ at the time point before the last time point when the studies were terminated. The age for all mice used in this study is from 8 weeks old. The mice were housed in a specific pathogen-free barrier environment (around 20 °C with 40% humidity and a 12 h day–night cycle). The experimental and control animals were bred separately.

### Materials
RAW 264.7 cells and B16F10 cells were purchased from the National Infrastructure of Biomedical Cell Line Resource (Beijing, China). The construct expressing luciferase (Luci) was introduced into B16F10 cells by lentiviral transduction for bioluminescence imaging. Cell lines were checked free of mycoplasma contamination by PCR and culture. The identity of the cell lines was authenticated with STR profiling in 2019. Fetal bovine serum (FBS), Dulbecco's modified Eagle medium (DMEM), and Roswell Park Memorial Institute 1640 (RPMI 1640) were purchased from Sigma (USA). Cell-Counting Kit 8 (CCK8) were purchased from Yeasen (Shanghai, China). Transwell system was purchased from Corning (USA). 1,1'-dioctadecyl-3,3,3',3'-tetramethylindocarbocyanine perchlorate (DiI) dye was purchased from Beyotime Biotechnology (China). Cell tracker CM-DiI was pur-chased from Yeasen Biotechnology (China). CellTrace™ CFSE Cell Proliferation Kit was purchased from Thermo Fisher Scientific (USA). Anti-CD11b-APC-Cy7 (Cat #101226, clone number: M1/70, 0.2 mg/ml), anti-F4/80-FITC (Cat #123107, clone number: BM8, 0.5 mg/ml), anti-CD45-Brilliant Violet 421 (Cat #103134, clone number: 30-F11, 0.2 mg/ml), anti-CD206-PE (Cat #141706, clone number: C068C2, 0.2 mg/ml), anti-CD86-APC (Cat #105011, clone number: GL-1, 0.2 mg/ml), anti-CD206-APC (Cat #142707, clone number: C068C2, 0.2 mg/ml), anti-CD8-PE (Cat #100708, clone number: 53-6.7, 0.2 mg/ml), anti-CD4-APC (Cat #100412, clone number: GK1.5, 0.2 mg/ml), anti-CD3-APC-Cy7 (Cat #100221, clone number: 17A2, 0.2 mg/ml) and 7-AAD Viability Staining Solution (Cat #420403, 5 µl of 7-AAD per million cells) were purchased from BioLegend (USA). Anti-mouse CD86 Rabbit mAb (Cat #19589, clone number: E5W6H, dilution: 1:200), anti-mouse CD206 Rabbit mAb (Cat #24595, clone number: E6T5J, dilution: 1:200) and anti-mouse F4/80 Rabbit mAb (Cat #70076, clone number: D2S9R, dilution: 1:200) were purchased from Cell Signaling Technology (USA). D-luciferin-K⁺ salt bioluminescent sub-strate and macrophage colony-stimulating factor (M-CSF) were pur-chased from Sigma–Aldrich (USA). TNF-α Elisa Kit, IFN-γ Elisa Kit, IL-4 Elisa Kit, and IL-10 Elisa Kit were purchased from Multi Sciences (China). RNA Isolation Mini kit, HiScript II Q RT Super Mix, and SYBR-Green qPCR Master mix were purchased from Vazyme (China). InVivoMAb anti-mouse IFN-γ and InVivoMab rat IgG1 isotype control (anti-HRP) were purchased from Bio X Cell (USA).

### Cell culture
B16F10 cells were cultured in RPMI 1640 containing 10% FBS. RAW 264.7 cells were cultured in DMEM containing 10% FBS. Bone marrow-derived macrophages (BMDMs) were generated from the bone mar-row of C57BL/6 mice. Briefly, the tibia and femur were isolated from 6–8-week C57BL/6 mice. Bones were kept on ice and rinsed in a sterile dish. In a sterile environment, the ends of each bone were transected and the marrow cavity was flushed with culture medium using a sterile syringe. After lysis of the red blood cells, harvested cells were washed and cultured in DMEM containing 10% FBS and 25 ng/mL M-CSF, and allowed to differentiate into macrophages for 7 days. All cells were cultured in a humidified atmosphere of 5% CO₂ and 95% air at 37 °C.

### Overall survival analysis
The Gene Expression Profiling Interactive Analysis (GEPIA2) browser (http://gepia2.cancer-pku.cn) is a web-based tool for analyzing the

data provided by TCGA and genotype-tissue expression[49]. We analyzed the correlation of the expression of *CD86* or *IFNG* with prognostic outcome in melanoma using the GEPIA2 browser, as well as the correlations between *CD86* and *IFNG*.

## Construction and characterization of genetically engineered macrophages

To develop engineered macrophages which can secrete IFN-γ through wireless remote control, RAW 264.7 cells or BMDMs were transduced by lentivirus encoding heat-inducible dCas9 fused with 10 copies of GCN4, and scFv fused with p65-HSF1 transcriptional activator and sgRNA, respectively[37]. The transduced cells were selected by G418 and puromycin to obtain the engineered macrophages. The targets to activate *Ifng* gene were designed by online tool platform http://chopchop.cbu.uib.no/[50]. Primers and oligos used in this study are listed in Supplementary Table 1. Three annealed oligos were cloned into U6-sgRNA-CMV-scFv-p65-HSF1-P2A-Puro plasmids. To construct HSP70-dCas9-10×GCN4-P2A-Neo plasmids, HSP70 promoter core sequence and dCas9 sequence were amplified by PCR, and plasmids were constructed by homologous recombination[23]. Other plasmids were constructed similarly using the same molecular cloning methods.

To develop engineered macrophages which carrying EGFP reporter, RAW264.7 cells were transduced by lentivirus encoding EGFP reporter driven by HSP70 promoter. The transduced cells were selected by puromycin to obtain the engineered macrophages.

## Cell viability assay

Cells (BMDMs, RAW 264.7, B16F10) were seeded in 96-well plates overnight. Then, cells were heated for different time or at different temperature using iWarm (place iWarm on the bottom of the plate). After 24 h, the supernatant in the plates was removed, CCK8 (10 μl/well in 90 μl fresh medium) was subsequently added to the plates followed by incubation at 37 °C for 1 h. After shaking the plates for 60 s, absorbance values at 450 nm of each well were recorded by a microplate reader.

## Cell viability of B16F10 cells in co-cultured assay

$2 \times 10^4$ eRAW 264.7 or eBMDM were seeded into the upper chamber of 24-well transwell plates overnight, while $5 \times 10^3$ B16F10 cells were seeded into the lower chamber in 24-well transwell system overnight. The system was heated at 42 °C for 30 min, and then cultured at 37 °C for 48 h. CCK8 (50 μl/well in 450 μl fresh medium) was subsequently added to the plates followed by incubation at 37 °C for 1 h. After shaking the plates for 60 s, the supernatants in plates were transferred into 96-wells plate. Absorbance values at 450 nm of each well were recorded by microplate reader.

## The ON/OFF ratio, ON/OFF kinetics, and leaky effect of the gene circuit

RAW264.7 cells were transfected with HSP70-EGFP plasmids to obtain engineered RAW264.7 which carrying EGFP reporter. eRAW264.7 cells were then heated at different temperature for 30 min using iWarm. The positive EGFP cells were evaluated 24 h after the heat shock and quantitative analysis of EGFP fluorescence by ImageJ. Then, the ON/OFF ratio and leaky effect of the gene circuit were calculated though the mean fluorescence intensity. EGFP expression 24 h after heat shock at 42 °C for 30 min at day 0, day 3 and day 6, respectively. The EGFP-positive eRAW264.7 cells were further quantified by ImageJ. And the time-dependent HS-mediated OFF kinetics of transgene expression was calculated though the mean fluorescence intensity.

## The mean number of integrated lentivirus copies per cell

To calculate the copy number, real-time quantitative PCR (RT-qPCR) was used to determine the mean number of integrated lentivirus copies per cell. In details, Woodchuck Hepatitis Virus (WHV) Post-

transcriptional Regulatory Element (WPRE) sequence, which is a common regulatory element in recombinant lentiviruses and does not normally exist in animal cells, was be used as a quantitative marker after the insertion of the recombinant lentivirus into the target cell genome, while the single-copy albumin (Alb) gene was used as the reference gene. The genome of lentivirus-infected cells and $10^4$–$10^7$ copies of standard sample were used as templates for quantitative PCR, and the standard curve was drawn according to the Ct value and copy number of the standard sample. The Ct value of the tested sample was substituted into the standard curve, and the copy number of WPRE and Alb of the tested sample was calculated. The mean number of integrated lentivirus copies per cell was calculated with the following formula: The mean number of integrated lentivirus copies per cell = (the mean number of WPRE gene copies/ the mean number of Alb gene copies) × 2.

## In vitro flow cytometry analysis

eRAW 264.7 or eBMDM were seeded into the upper chamber of 6-well transwell plates overnight. The next day, cells were heated at 42 °C for 30 min, and then cultured at 37 °C for 24 h. Then, the cells were collected and stained with fluorescence-labeled CD86. The samples were run on a BD LSRFortessa flow cytometer.

## In vitro phagocytosis assays of eMac

Before phagocytosis assay, eBMDM or eRAW264.7 were heated at 42 °C for 30 min, and then cultured at 37 °C. After 24 h, eBMDM or eRAW264.7 were collected and stained with DiI or CM-DiI according to the protocol of Cell Plasma Membrane Staining Kit with DiI. At the same time, B16F10 were collected and stained with CFSE according to the protocol of the Cell Trace CFSE Cell Proliferation Kit. Then, $2 \times 10^5$ DiI-labeled eMac were co-cultured with $2 \times 10^5$ CFSE-labeled B16F10 cells in 6-well plate in medium of cultured eBMDM or eRAW264.7. After 24 h, the cells were collected and analyzed by BD LSRFortessa flow cytometer. The phagocytosis ratio was evaluated by the percentage of CFSE-positive cells in DiI-positive cells.

## In vitro chemotaxis assays of eMac

$2 \times 10^4$ eRAW 264.7 or eBMDM and RAW 264.7 or BMDM were seeded into the upper chamber of 24-well transwell plates overnight, while $5 \times 10^4$ B16F10 cells were seeded into the lower chamber in 24-well transwell system overnight, the lower chamber with only culture medium (CM) served as the control group. The system was cultured at 37 °C for 48 h. eRAW 264.7 or eBMDM and RAW 264.7 or BMDM were washed and fixed with paraformaldehyde, and then stained with crystal violet. The chemotaxis was evaluated by the number of cells that migrates through the semipermeable membrane[51].

## RNA extraction, cDNA synthesis, and real-time PCR

Total RNA was extracted from various stages of differentiated cells using RNA Isolation Mini kit according to the manufacturer's protocol, and 1 μg RNA was reversely transcribed to cDNA with HiScript II Q RT Super Mix according to the manufacturer's protocol. Gene expression was analyzed with SYBR-Green qPCR Master mix using Thermo ABI QuantStudio 6. Primers used in this study are listed in Supplementary Table 1.

## In vitro cytokine concentrations analysis

The cytokine concentrations of mouse IFN-γ, IL-10, IL-4, and TNF-α were determined from the co-cultured supernatant of eBMDM or eRAW264.7 with B16F10 cells. All experiments were performed using corresponding ELISA Kits according to the manufacturer's instructions.

## In vivo tumor models and treatments

C57BL/6 mice (8 weeks old, male and female) (stock number: 00013) were purchased from Hangzhou Ziyuan Laboratory Animal Technology Co. LTD (Zhejiang, China). For the B16F10 tumor model, $1 \times 10^6$

luciferase-expressing B16F10 cells were subcutaneously injected into C57BL/6 mice. 7 days later, when tumors reached a size of about 50–80 mm³, the mice were randomly divided and intravenously injected with PBS or $5 \times 10^6$ eBMDM or eRAW264.7 every ten days. The day after the injection, tumors on the back of mice were heated at 42 °C by iWarm for 30 min or intravenously injected with IFN-γ every three days for 3 weeks. Tumor volume and body weight were measured every three days until the end of the experiment. Tumor growth was also monitored by in vivo luciferase using bioluminescence imaging system (IVIS Spectrum, PerkinElmer). Some of the mice were exposure to carbon dioxide to euthanasia on day 21, the tumor tissues, peripheral blood, skin tissue, and the major organ of mice were collected and analyzed. The remaining mice were used for the survival study. Mice were exposure to carbon dioxide to euthanasia when the volume of the tumor exceeded 2000 mm³ or when the mice became moribund. Survival was evaluated from the first day of implantation until day 60.

For the B16F10 tumor metastasis model, $1 \times 10^6$ luciferase-expressing B16F10 cells were subcutaneously injected into C57BL/6 mice. 7 days later when tumors reached a size of about 50–80 mm³, the mice were intravenously injected with $1 \times 10^6$ luciferase-expressing B16F10 cells. The next day, the mice were randomly divided and intravenously injected with PBS or $5 \times 10^6$ eBMDM or eRAW264.7. The day after the injection, tumors on the back of mice were heated at 42 °C by iWarm for 30 min every three days. After 3 weeks the mice were exposure to carbon dioxide to euthanasia and all the lungs were harvested, photographed, fixed, and sections were taken for H&E staining. The metastasis nudes were counted to evaluate the anti-metastasis effect. Lung metastasis was also detected by ex vivo luciferase based small animal live fluorescence image analysis system Caliper IVIS Spectrum.

For macrophage depletion model, $1 \times 10^6$ luciferase-expressing B16F10 cells were subcutaneously injected into C57BL/6 mice. 7 days later when tumors reached a size of about 50–80 mm³, the mice were randomly divided and intravenously injected with Clo (100 µl/10 g body weight, 5 mg/ml) at day 1, day 7 and day 13[52]. And then the mice were intravenously injected with PBS or $5 \times 10^6$ eBMDM or eRAW264.7 at day 2, day 8 and day 14, followed by using iWarm at 42 °C for 30 min at at day 3, day 5, day 9, day 11 and day 15. At day 16, the mice were exposure to carbon dioxide to euthanasia and the tumor tissues were collected and analyzed.

### Immunofluorescence

The tumor samples, which had been fixed in formalin and treated with paraffin, were used for immunofluorescence. The paraffin sections were first deparaffinized, rehydrated, treated with buffer for antigen retrieval, covered with blocking buffer, and then stained with fluorescence-labeled antibodies: F4/80, CD206, CD86, and DAPI was used to label the nuclear. After that, the sections were subjected to investigation by confocal microscopy Olympus FV3000.

### In vivo toxicity evaluation

Blood was collected and centrifuged from the tumor-bearing mice to test the levels of white blood cell, lymphocyte, monocyte, total protein, albumin, globulin, ALT (alanine aminotransferase), AST (aspartate aminotransferase), AST/ALT, uric acid and blood urea nitrogen in serum. For H&E staining, subcutaneous tumors, skin tissue, and other organs were dislodged and fixed in 4% paraformaldehyde.

### In vivo eMac survival assays

For in vivo eBMDM or eRAW264.7 survival assays, cells were labeled with DiI dye at 5 µM for 10 min at 37 °C before being injected to mice, and the mice were then monitored by a small animal live fluorescence image analysis system Caliper IVIS Spectrum for 10 days.

### In vivo flow cytometry analysis

The tumor tissues were enriched from mice, cut into small pieces, and digested in cell culture media supplementing with DNases (0.2 mg/ml, Sigma–Aldrich), collagenase D (1 mg/ml), and hyaluronidase (0.1 mg/ml) at 37 °C for 30 min, and were then filtered with 70-µm cell strainers. The cells were stained with fluorescence-labeled antibodies: CD45, CD3, CD4, CD8, CD11b, F4/80, CD206 and CD86. 7-AAD Viability Staining Solution was employed to exclude the dead cells. The samples were run on a BD LSRFortessa flow cytometer. Gating strategy of CD8⁺ T cells within CD45⁺ cells or CD3⁺ T cells and gating strategy of M1 and M2 macrophages were in Supplementary Fig. 23.

### Analysis of in vivo cytokine concentrations

The intratumor levels of IFN-γ, IL-10, IL-4, and TNF-α were measured by using corresponding ELISA Kits according to the manufacturer's instructions.

### Statistics & reproducibility

All data and figures in this paper were analyzed and plotted by GraphPad prism 8.0, FlowJo V10, ImageJ 1.53, Adobe Photoshop 2023, and Adobe Illustrator 2023. All results were calculated by expressing mean ± standard deviation (S.D.). Biological replicates were used in all experiments unless otherwise stated. Paired or unpaired two-tailed Students' $t$-test was used for comparison of two groups. One-way analysis of variance (ANOVA) with a Tukey's post-hoc test or Dunnett's multiple comparison or Sidak's multiple comparisons was used when more than two groups were compared. The $p$-value less than 0.05 was considered significant ($*p < 0.05$, $**p < 0.01$, $***p < 0.001$, $****p < 0.0001$). For all the experiments, reproducibility is stated in figure legends. No data were excluded.

### Schematic illustrations

Schematic illustrations were created with Biorender.com, Adobe Photoshop, and Adobe Illustrator.

### Reporting summary

Further information on research design is available in the Nature Portfolio Reporting Summary linked to this article.

## Data availability

All data generated or analyzed during this study are included in this article and its supplemental materials. The correlation of gene expression with prognostic outcome were analyzed by GEPIA2 browser (http://gepia2.cancer-pku.cn). Source data are provided with this paper.

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

## Acknowledgements

This work was supported by National Natural Science Foundation of
China (82202975, Y.X., 82073779, Y.P., 81573003, D.L.), National Key
R&D Program of China (2021YFA0909900, Y.P.), Natural Science Foun-
dation of Zhejiang Province (LQ21H180010, Y.X.). We acknowledge Dr.
Hui Yang from Institute of Neuroscience, State Key Laboratory of Neu-
roscience, Chinese Academy of Sciences for providing the plasmids
ofEF1α-scFv-p65-HSF1-T2A-EGFP-WPRE-PolyA and dSV40-NLS-dCas9-
HA-NLS-NLS-10xGCN4. We thank for the technical support by School of
Medicine, College of Pharmaceutical Sciences and Life Sciences Insti-
tute, Zhejiang University.

## Author contributions

Y.P. designed research; Y.X., X.Y., and S.D. performed research; Y.X.,
X.Y., D.L., S.D., and Y.P. analyzed data; and Y.P. wrote the paper.

## Competing interests

The authors declare no competing interests.

## Additional information

**Supplementary information** The online version contains
supplementary material available at

Yuan Ping.

**Peer review information** *Nature Communications* thanks Anders Etzer-
odt and the other, anonymous, reviewer(s) for their contribution to the
peer review of this work. A peer review file is available.

