## [Peer Review File · Nature Communications]

Proinflammatory polarization of engineered heat-inducible macrophages reprogram the tumour immune microenvironment during cancer immunotherapyREVIEWER COMMENTS

Reviewer #1 (Remarks to the Author):

The authors developed an engineered macrophage which can respond to mild temperature elevation and locally secrete IFN- γ in the tumor tissue controlled by an iWarm device. Thus, self-polarization of eMacs and re-polarization of TAMs in TME can be accomplished. All data indicated that the strategy can dramatically inhibit the tumor growth and lung metastasis. Overall, the story is very interesting and the study was well organized. However, I still have some concerns.

1. In my opinion, the primary objective of this study is to evaluate if the eMacs is well engineered for cancer immunotherapy. As eMacs is thermo-sensitive, the authors also designed a wireless remote iWarm device to control the local temperature. However, I think that iWarm is just a gimmick in this study. Actually, the mild temperature elevation effect can be achieved by some simpler methods when compared to iWarm. The authors should concentrate their attention on evaluating the performance of eMacs rather than designing a wireless remote heating device. I suggest that the contents about iWarm should be removed from the main body of the manuscript. And it is unnecessary to emphasize the significance of the wireless remote temperature control in this study.

2. The term of "wireless-controlled eMacs" is not appropriate. Actually, the iWarm and the heat-shock are wireless-controlled. The eMacs are thermal-responsive.

3. Fig. 1d shows the prototype of iWarm that is readily wrapped onto the mouse body through a tailored overcoat. It is not unpractical to let the mouse wear the iWarm-containing overcoat all the time. Thus, the mice should be anaesthetized and wrapped by the overcoat before every heating. It is too complicated.

4. More details about the experiments should be given. (1) the cell density in co-culture assay, phagocytosis assay and chemotaxis assay, which may have significant influence on the results. (2) how to heat cells in vitro?

5. The author should not only investigate the effect of temperature on eMacs, but also investigate on the effect of temperature on BMDM. (1) the cytotoxicity of heat shock towards BMDM and eMacs should be investigated, (2) whether the temperature also could lead to the repolarization of BMDM in vitro or TAMs in vivo.

6. What subtype of BMDMs was used for experiments?

7. Could the eBMDM be polarized without heat shock? Since only 50% of B16F10 cells are alive after co-cultured with eBMDM in Fig 3i. The phenotype of eBMDM should be checked.

8. Fig 6i is the same as Fig 6d, which cannot illustrate how to establish the B16F10 tumor metastasis model.

9. A min and max bar and data analysis of the fluorescence intensity in Fig 6e are suggested to be supplemented.

10. The scale bar in Fig S9 should be checked.

11. Fig 3m and Fig S3i are not clear.

12. Line 174, the "." behind promoter should be omitted.

Reviewer #2 (Remarks to the Author):

This article by Xue et al. presents data to support the engineering of macrophages for cancer immunotherapy. They devise a thermal sensitive promoter based on the heat shock system to control the expression of IFN- γ via a dead Cas9 construct. The idea is to polarize macrophages to the inflammatory M1 state (both engineered MACs (eMACs) as well as tumor associated macrophages). They devised a wearable heating device called iWARM and show that it can be controlled remotely by a cell phone to heat tissues (chicken breast) of up to 1 cm thickness to 42 degrees centigrade. Overall, there is a need for increased spatial and temporal control of cellular immunotherapies. This work contributes to the field by showing that macrophages can be engineered and triggered noninvasively to produce localized IFN- γ . The data to support this premise, in its current form, however, needs more work and is open to interpretation.

Major Concerns

- The data in Figure 2 shows RAW macrophage cells responding to heat. However, these experiments were done by plasmid transfection and as a result, each cell has many copies of the plasmid – more so that what can be obtained for a lenti or retroviral system. As the FDA limits the number of integration sites (<5) per cell to prevent insertional mutagenesis, it is not clear the ON/OFF ratio, ON/OFF kinetics, and background leakiness of the circuit. The plasmids will also be lost eventually; it is not clear whether that affects the repeat heating data as well.
- In Figure 3a,b,c., the eBMDM was heat treated for 30 minutes yet the production of IFN-g continues to increase to 72 hrs. The heat shock response is transient, and other reports have shown that protein expression decreases 18-24 hrs after heat shock when the transgene mRNA has been degraded. Why is it that the IFN-g production does not come back down, or even plateau? One would really want a reversible control system given the toxicity of IFN-gamma.
- It is not clear what figure 3k was gated on, as the data should show double positive macrophages to support phagocytosis? The data only shows the FITC channel. A co-culture experiment of macrophages and tumors cells in the presence of a phagocytosis inhibitor is required to claim that the mechanism is phagocytosis. A co-culture experiment is also required to show that the DiI lipophilic dye used to stain macrophages does not spontaneously transfer to the membrane of the B16 tumor cells.
- Figure 3m is very difficult to interpret with poor image quality and resolution. What evidence is there that phagocytosis of a target cell membrane would lead to presentation of those molecules on the macrophage cell surface? The data presented is a single field of view with no quantification.
- Please change the use of eMac in the manuscript. There is a significant difference if the experiments were carried out with bone marrow derived macrophages or RAW cells. The latter is not translatable. If eBMDM were used, use eBMDM in the text and figures. If engineered RAW cells were used, indicate as eRAW cells (or equivalent) in the text and figures.
- Figure 4a,b,c require biological replicates and statistics. One cannot draw any conclusion based on the luminescent pictures themselves.
- Interpretation of data is SFig S4 is generous. It could be that the RAW cells are alive but that the lipophilic dye is degraded, or that the RAW cells have undergone multiple rounds of cell division and dilute the membrane dye. Please repeat using a genetically encoded luminescent reporter.
- For Figure s6, the experimental timeline is not clear. When were tumors heated and when was the IFNg administered?
- Figure 5j, k. are uninterpretable and unquantified IF images.
- Figure 6b is impossible to interpret. There is no validation that the CLo really depletes TAMs. Figure 6e requires biological replicates with statistics.
- Figure 7g does not evidence that the intestines have been brought up to 42oC. The thermal photographs show that the surface temperature is 42C. The claims that inflammatory bowel disease treatment or Hirschsprung disease treatment can be addressed by this approach is not supported by the data.
- The claim that IFN-gamma production “was reversible and can be terminated simply through switching off the iWarm to avoid the unwanted side effects, such as cytokine release syndrome and immune effector cell-associated neurotoxicity syndrome that are commonly encountered in other cell therapies, such as CAR-T therapy” is not supported by the data. In fact, the authors show that the IFN-gamma levels do not come back down even after 72hrs after heat shock (Figure 3a,b,c)

Minor points

- Please remove subjective language throughout the paper, including but not limited to “Encouraged by these exciting results” page 6, “interestingly, emacs could migrate” page 7
- Please include citations of spatial / temporal control of dCas9 by light (Polstein et al. Nat. Chem. Biol. 2015), heat (Gamboa et al. ACS Chem Biol 2020), and small molecules (de Souza et al. Nat. Methods 2015).

Reviewer #3 (Remarks to the Author):

In the manuscript "Wireless Control of Engineered Heat-Inducible Macrophages for Cancer Immunotherapy" submitted by Yanan Xue et al. the authors propose a method for inducing anti-tumor immune responses using adoptive transfer of macrophages transfected with heat-inducible expression of IFN γ . Overall, the manuscript is well written and provides evidence that heat-inducible expression of IFN γ in adoptively transferred macrophage can induce anti-tumor responses in mice. However, a few questions remain to answered:

1. The authors state that eRAW264.7 cells could survive in tumor tissue for up to 10 days. How long did the eBMDMs remain in the tumor and healthy tissue? To better understand the proposed system, it would also be interesting to know the ratio of eBMDMs compared to host TAM
2. In fig 2, induction of IFN γ is measured at 37°C, 42°C, 45°C and 48°C. What level of induction is seen between 37°C-42°C?
3. Throughout the manuscript, the reader gets the impression that heat inducible expression of IFN γ in macrophages is safe although no data on toxicity are presented. Given the infiltration of eBMDMs in normal tissue as well, what would be the effect of slightly elevated body temperatures? This should be tested in an appropriate model (e.g. experimental endotoxemia model)
4. In figure 5 the authors claim that heat inducible expression of IFN γ drives an increase in CD4 and CD8 T cells. However, as the data only shows increase in CD4+ and CD8+ cells out of CD3+ T cells it is not possible to conclude that T cells are increased. Instead, the authors should as a minimum show the Frequency of CD4 and CD8 T cells out of live cells or preferentially, number of cells pr tissue weight. Also, in control mice, less than 40% of the CD3+ cells are CD4 or CD8, which makes the reader question the gating strategy used. As a minimum this should be available in the supporting information
5. In figure 6, the authors use clodronate liposomes to prove the effect of eBMDMs. However, clodronate Liposomes depletes all macrophages and the differences between clodronate treated eBMDMs + HS and eBMDMs + HS seem minor. To prove that the observed treatment effect is due heat inducible IFN γ expression, it would make more sense to include an IFN γ /IFN γ R blocking antibody

Point-by-Point Response to the Reviewers' Comments

(Manuscript ID: NCOMMS-22-52387A)

Reviewer 1

1. In my opinion, the primary objective of this study is to evaluate if the eMacs is well engineered for cancer immunotherapy. As eMacs is thermo-sensitive, the authors also designed a wireless remote iWarm device to control the local temperature. However, I think that iWarm is just a gimmick in this study. Actually, the mild temperature elevation effect can be achieved by some simpler methods when compared to iWarm. The authors should concentrate their attention on evaluating the performance of eMacs rather than designing a wireless remote heating device. I suggest that the contents about iWarm should be removed from the main body of the manuscript. And it is unnecessary to emphasize the significance of the wireless remote temperature control in this study.

Reply: Thanks for this suggestion. We agree the reviewer's suggestion to remove the contents regarding iWarm (in Figure 1d-g and Figure 7a-c) from the main body of the manuscript and move to the Supporting Information as Figure S1a-d. In the meantime, according to the editor's suggestion, we also removed Figure 7 on the human tolerability data which we believe unnecessary, and weakened the significance of the wireless remote control of temperature in the revised manuscript.

Fig. S1 Characterization of the iWarm. (a) The prototype of iWarm. (b) The locoregional temperature of the mouse back after turning on iWarm monitored by thermal imaging equipment. (c) The thickness of the chicken breast. (d) The temperature of contact surface (front side) and the non-contact surface (back side) before and after iWarm-mediated temperature elevation monitored by a thermal imaging camera.

2. The term of "wireless-controlled eMacs" is not appropriate. Actually, the iWarm and the heat-shock are wireless-controlled. The eMacs are thermal-responsive.

Reply: Thanks for the suggestion. We agree that iWarm and heat-shock are wireless-controlled, whereas eMacs are thermal-sensitive. Thus, we revised "wireless-controlled eMacs" as "thermo-responsive eMacs".

3. Fig. 1d shows the prototype of iWarm that is readily wrapped onto the mouse body through a tailored overcoat. It is not unpractical to let the mouse wear the iWarm-containing overcoat all the time. Thus, the mice should be anaesthetized and wrapped by the overcoat before every heating. It is too complicated.

Reply: Thanks for pointing out this issue. Fig. S1a (in the Supporting Information) is an illustration of thermal stimulation of mice with iWarm. Indeed, our studies on thermal stimulation by iWarm were carried out by anaesthetizing the mice first to avoid any movement of the mice, followed by wearing the iWarm-containing overcoat and elevating its temperature locally. Alternatively, the anesthetic procedures can be also simplified by immobilizing the mice to ensure the efficient transmission of heat from iWarm to the tumor. As this is a proof-of-concept study, the future translation on human makes it possible to wear the customized iWarm-

containing overcoat during the treatment, without any anesthetic procedures as in experimental animals.

Fig. S1 (a) The prototype of iWarm.

4. More details about the experiments should be given. (1) the cell density in co-culture assay, phagocytosis assay and chemotaxis assay, which may have significant influence on the results. (2) how to heat cells *in vitro*?

Reply: Thanks for the suggestion. We have supplemented the methods in the revised manuscript.

(1) Before phagocytosis assay, 2×10^5 DiI-labelled eMacs were co-cultured with 2×10^5 CFSE-labelled B16F10 cells in 6-well plate in medium of cultured eMacs.

(2) Before chemotaxis assay, 2×10^4 eRAW 264.7 or eBMDM and RAW 264.7 or BMDM were seeded into the upper chamber of 24-well transwell plates, while 5×10^4 B16F10 cells were seeded into the lower chamber in 24-well transwell system.

(3) Cells (BMDMs, RAW 264.7, B16F10) were heated using iWarm which was attached onto the bottom of the plate *in vitro*. The plate was pre-heated for 30 min to reach 42 °C before the heat activation assay.

The above information is supplemented in the Method section.

5. The author should not only investigate the effect of temperature on eMacs, but also investigate on the effect of temperature on BMDM. (1) the cytotoxicity of heat shock towards BMDM and eMacs should be investigated, (2) whether the temperature also could lead to the repolarization of BMDM *in vitro* or TAMs *in vivo*.

Reply: Thanks for the useful comments. As requested by the Reviewer, we have investigated the effect of temperature on BMDM using CCK8 after the indicated treatment in Figure S3c and S3f. The results showed that the incubation of BMDMs under 42 °C for 30 min (the condition for heat activation) merely affect their cell viability. However, experimental conditions (temperature and duration) beyond results in slight cytotoxicity.

Fig. S3 Cell viability of B16F10 or RAW264.7 or BMDM determined by CCK8 after wireless-controlled heating at different temperature for 30 min (a, b, c), or at 42 °C for different duration (d, e, f). Data are representative of independent experiments with similar results. Values for n represent biologically independent samples. Data are presented as mean \pm SD (n = 3). Statistical significance was calculated via one-way ANOVA with a Dunnett's multiple comparison tests in a-f. * $P < 0.05$; ** $P < 0.01$; *** $P < 0.001$; **** $P < 0.0001$. Source data are provided as a Source Data file.

In addition, the effect of temperature on BMDM phenotypes was also investigated, and we found that the temperature change could not alter the phenotype of BMDM (Figure S4a and S4b). Then, we investigated the effect of different temperature on M2 macrophage (IL4-treated BMDM) phenotype. Similarly, the temperature at 42 °C for 30 min did not alter the phenotype of M2 macrophage *in vitro* (Figure S4c and S4d). Furthermore, our results in Figure 4a-d and Figure S18a-d demonstrated that the temperature at 42 °C lasted for 30 min did not alter the phenotype of TAMs *in vivo*.

Fig. S4 Flow cytometry analysis of the M1 polarization (a) and the M2 polarization (b) of BMDM after the heating at different temperature for 30 min. Flow cytometry analysis of the M1 polarization (c) and the M2 polarization (d) of IL4-treated BMDM after the heating at different temperature for 30 min. CD86⁺ is the marker of M1 macrophages, while CD206⁺ is the marker of M2 macrophages.

Fig. 4 RT-qPCR analysis of M1 (a) and M2 (b) macrophages markers in tumor tissues after the indicated treatment. (c) Flow cytometry analysis of the polarization of macrophages in tumor tissues after the indicated treatment. CD86⁺ is the marker of M1 macrophages, while CD206⁺ is the marker of M2 macrophages. (d) Quantitative analysis of the M1/M2 ratio of macrophages in (c).

Fig. S18 RT-qPCR analysis of M1 (a) and M2 (b) macrophages markers in tumor tissues after the indicated treatment. (c) Flow cytometry analysis of the polarization of macrophages in tumor tissues after the indicated treatment. CD86⁺ is the marker of M1 macrophages, while CD206⁺ is the marker of M2 macrophages. (d) Quantitative analysis of M1/M2 ratio of macrophages in (c).

6. What subtype of BMDMs was used for experiments?

Reply: Thanks for the comment. In this study, BMDMs were generated from the bone marrow of C57BL/6 mice and differentiated into M0 macrophages by macrophage colony stimulating factor (M-CSF).

7. Could the eBMDM could be polarized without heat shock? Since only 50% of B16F10 cells alive after co-cultured with eBMDM in Fig3i. The phenotype of eBMDM should be checked.

Reply: Thank you for the insightful comment. To investigate the phenotype of eBMDM in Transwell system, we cultured B16F10 in the upper chamber of Transwell system, and eBMDM cells were separately cultured in the lower chamber. Following the heat shock at 42 °C for 30 min, both cells were continued to culture for 24 h, and the polarization of M1 macrophages by flow cytometry (Figure S7). The results showed that compared with BMDM cultured alone (NC group), control group, HS group and eBMDM group (without heat shock) were slightly polarized into M1 phenotype, and engineered BMDM after heat shock could significantly polarize to M1 (Figure S7).

The above information has been supplemented in the revised manuscript.

Fig. S7 Flow cytometric analysis of CD86⁺ macrophages in eBMDM after the indicated treatment.

8. Fig 6i is the same to Fig 6d, which cannot illustrate how to establish the B16F10 tumor metastasis model.

Reply: Thank you for the useful comments. We conjecture that the Reviewer may question Figure 4d and 4i (as shown below) rather than Figure 6d and 6i. The illustration between Figures 4d and 4i is slightly different on day 7 (highlighted in the red square in Figure 4i), when B16F10 cells were administrated into mice by tail vein to establish the B16F10 tumor metastasis model.

9. A min and max bar and data analysis of the fluorescence intensity in Fig6e are suggested to be supplemented.

Reply: Thank you for the comment. We conjecture that the Reviewer may have questioned Figure 6c rather than Figure 6e. We have supplemented the min and max bar of the bioluminescence intensity in Figure 6c (shown as Figure 5b in the revised manuscript).

Fig. 5 (b) *In vivo* bioluminescence images of mice after the specified treatment at day 16.

10. The scale bar in Fig S9 should be checked.

Reply: Thanks for the reminder and we are sorry for this omission. We have corrected this issue in the revised manuscript.

Fig. S14 (c) H&E staining of mouse skin slice at different time points after using iWarm-enabled locoregional hyperthermia at 42 °C for 30 min every day.

11. Fig 3m and Fig S3i are not clear.

Reply: Thanks for pointing out this issue. As requested by the reviewer, we have zoomed in on the picture in Figure 3m and Figure S3i (shown as Figure 2m and Figure S6i in the revised manuscript) to make them clear.

Fig. 2 (m) Fluorescence images of the phagocytosis by BMDMs. B16F10 cells were labeled with CFSE (green), the cell membrane of eBMDM was labeled with Dil (red), and the nuclei of both cells were stained with DAPI (blue). The white arrows point to the B16F10 cell phagocytized by macrophages.

Fig. S6 (i) Fluorescence images of the phagocytosis by RAW264.7. B16F10 cells were labeled with CFSE, the cell membrane of RAW264.7 was labeled with DiI, and the nuclei of both cells were stained with DAPI. The white arrows point to the B16F10 cell phagocytized by macrophages.

12. Line 174, the "." behind promoter should be omitted.

Reply: Thanks for the reminder and we are sorry for this omission. We have corrected this issue in the revised manuscript.

Reviewer 2

Major Concerns

1. The data in Figure 2 shows RAW macrophage cells responding to heat. However, these experiments were done by plasmid transfection and as a result, each cell has many copies of the plasmid – more so that what can be obtained for a lenti or retroviral system. As the FDA limits the number of integration sites (<5) per cell to prevent insertional mutagenesis, it is not clear the ON/OFF ratio, ON/OFF kinetics, and background leakiness of the circuit. The plasmids will also be lost eventually; it is not clear whether that affects the repeat heating data as well.

Reply: Thanks for the insightful comments. To calculate the copy number, real-time quantitative PCR (RT-qPCR) was used to determine the mean number of integrated lentivirus copies per cell. In details, Woodchuck Hepatitis Virus (WHV) Post-transcriptional Regulatory Element (WPRE) sequence, which is a common regulatory element in recombinant lentiviruses and does not normally exist in animal cells, was used as a quantitative marker after the insertion of the recombinant lentivirus into the target cell genome, while the single-copy albumin (Alb) gene was used as the reference gene. The genome of lentivirus infected cells and 10^4 - 10^7 copies of standard sample were used as templates for quantitative PCR, and the standard curve was drawn according to the Ct value and copy number of the standard sample, $Y = -4.122 \times X + 49.58$ for Alb and $Y = -4.329 \times X + 50.39$ for WPRE. The Ct value of the tested sample (25.34 for Alb and 17.47 for WPRE) was substituted into the standard curve, and the copy number worked out to be 7.6 for WPRE and 5.89 for Alb of the tested sample. The mean number of integrated lentivirus copies per cell was calculated with the following formula: The mean number of integrated lentivirus copies per cell = (the mean number of WPRE gene copies / The mean number of Alb gene copies) \times 2. By calculation, we finally obtained an average copy number of integrated lentivirus per cell of 2.58.

In Figure 2 (shown as Figure 1 in the revised manuscript), we engineered RAW264.7 with HSP70-EGFP reporter through the lentiviral transduction rather than non-viral transfection, thus making eBMDM longer periods of transgene expression after heat activation. Our results showed that once EGFP expression was continuous for at least 9 days by heat activation every three day (Fig.1i). The gene circuit of eBMDM is highly sensitive to heat and is reversible, with distinct ON/OFF kinetics (Fig.1i-j). Furthermore, we analyzed the ON/OFF ratio and the leaky effect of the gene circuit. When the iWarm was turned on, EGFP expression was significantly increased by about 41.7-fold (Fig.S2a), and the background leakiness of the gene circuit was only 2.4% (Fig.S2b).

Fig. 1 (i) EGFP expression 24 h after heat shock at 42 °C for 30 min at day 0, day 3 and day 6, respectively. (j) Illumination time-dependent HS-mediated OFF kinetics of transgene expression.

Fig. S2 The ON/OFF ratio (a) and the leaky effect (b) of the gene circuit calculated according to Fig. 1e and 1f. Data are presented as mean \pm SD ($n = 3$). Statistical significance was calculated via one-way analysis of variance (ANOVA) with a Tukey post-hoc test. * $P < 0.5$; ** $P < 0.1$; *** $P < 0.01$; **** $P < 0.001$.

2. In Figure 3a,b,c (shown as Figure 2a-c in the revised manuscript and also below), the eBMDM was heat treated for 30 minutes yet the production of IFN-g continues to increase to 72 hrs. The heat shock response is transient, and other reports have shown that protein expression decreases 18-24 hrs after heat shock when the transgene mRNA has been degraded. Why is it that the IFN-g production does not come back down, or even plateau? One would really want a reversible control system given the toxicity of IFN-gamma.

Reply: Thanks for the comment. In the first, though some reports have shown that protein expression decreases 18-24 h after the heat shock due to the degradation of transgene mRNA, the expression of IFN- γ driven by heat-shock promoter was not transient due to the integration of transgene into the host genome by lentiviral transduction, thereby making eBMDM longer periods of transgene expression. The secreted IFN- γ binds to its receptor (the heterodimeric IFNGR1/IFNGR2 receptor complex) and induces the activation of transcription elements (such as members of the signal transducer and activator of transcription (STAT) family, mainly STAT4¹, T-box transcription factor (T-bet)², activator protein 1 (AP-1)³, or Eomes⁴), which further drives the production of IFN- γ . In addition, IFN- γ may also stimulate antigen-presenting cells to secrete IL-12, which triggers the re-activation of the IFN- γ production cycle. This phenomenon is known as the positive feedback loop of IFN- γ synthesis⁵. Thus, there is a continuous increase in IFN- γ production. However, after stimulation, the increased IFN- γ gradually decreased with the time (Fig. 2c).

Fig. 2 (a) Illustration of engineering process of BMDM and the polarization of eBMDM by wireless-controlled secretion of IFN- γ . (b) The secretion of IFN- γ by eBMDM after HS at 42 °C for 30 min. (c) The increased IFN- γ by eBMDM after HS at 42 °C for 30 min at day 1 and day 3, respectively. The green arrow in (b) and (c) refers to the time point of HS.

3. It is not clear what Figure 3k was gated on, as the data should show double positive macrophages to support phagocytosis? The data only shows the FITC channel. A co-culture experiment of macrophages and tumor cells in the presence of a phagocytosis inhibitor is required to claim that the mechanism is phagocytosis. A co-culture experiment is also required to show that the DiI lipophilic dye used to stain macrophages does not spontaneously transfer to the membrane of the B16 tumor cells.

Reply: Thanks for the useful comments. In the phagocytosis assay, eRAW264.7 and eBMDM cells were labeled with DiI (gate on PE) and B16F10 cells were labeled with CFSE (gate on FITC), which were analyzed by flow cytometry. Firstly, eRAW264.7 or eBMDM cells were first gated by PE to obtain PE-positive macrophages, in which FITC-positive cells were further gated to represent the macrophage population that

engulfs the B16F10 cells. Thus, Figure 3k (shown as Figure 2k in the revised manuscript) showed the FITC-positive cells within the PE-positive cells.

Fig. 2 (k) Flow cytometry analysis of the phagocytosis of BMDMs after the specified treatment.

As requested by the reviewer, to detect whether the DiI lipophilic dye used to stain macrophages would have leaky effect, we first co-cultured DiI-labeled RAW264.7 cells with unlabeled RAW264.7 cells in a ratio of 1:3 for 24 h. In the meantime, we cultured DiI-labeled RAW264.7 cells and RAW264.7 cells for 24 h separately, and then mixed in a ratio of 1:3 as the control group. If the DiI lipophilic dye used to stain macrophages spontaneously transfer to the unlabeled macrophage membrane, the ratio of PE-positive cells will increase as compared with the control group, as reflected by flow cytometry. In our experiment, we found there was no significant difference between co-culture group and separately cultured control group, suggesting that the leaky effect is very low (Fig. R1).

Fig. R1 (a) Flow cytometric analysis of DiI-positive RAW264.7. (b) Quantitative analysis of DiI-positive cells in (a). Data are representative of three independent experiments with similar results. Values for n represent biologically independent samples. Data are presented as mean \pm SD (n = 3). Statistical significance was calculated via one-way ANOVA with a Tukey's multiple comparison test in c. * $P < 0.5$; ** $P < 0.1$; *** $P < 0.01$; **** $P < 0.001$. Source data are provided as a Source Data file.

The phagocytic function of macrophages mainly depends on the activated "eat me" receptors and the inhibitory "don't eat me" receptors on the surface of macrophages. The recognized "Don't eat me" receptor on the surface of macrophages is SIRP α , which interacts with its ligand CD47 on tumor cells. SIRP α undergoes tyrosine phosphorylation and recruits the protein tyrosine phosphatases SHP-1 and SHP-2. These phosphatases inhibit the ability of prophagocytic receptors to trigger phagocytosis when ligands are present on tumor cells. When the interaction between SIRP α with CD47 is prevented, the prophagocytic receptors are able to induce productive activating signals, which lead to phagocytosis^{6,7}. As requested by the reviewer, to verify the phagocytosis of tumor cells by macrophages, we detected the capability of BMDM for phagocytosis in a co-culture system, in which CM-DiI-labeled BMDM were cultured with CD47-knockdown B16F10 cells. The result indicated that the knockdown of CD47 in B16F10 cells by siRNA significantly increased the phagocytic capability of BMDM (Fig. S8). This suggests that phagocytosis is an important mechanism by which macrophages exert anti-tumor effects.

Fig. S8 Flow cytometry analysis of the phagocytosis of BMDM in a co-culture system, in which DiI-labeled BMDM were cultured with B16F10 cells or CD47-knockdown B16F10 cells.

4. Figure 3m is very difficult to interpret with poor image quality and resolution. What evidence is there that phagocytosis of a target cell membrane would lead to presentation of those molecules on the macrophage cell surface? The data presented is a single field of view with no quantification.

Reply: Thank you for highlighting this point. In the phagocytosis assay, eBMDM were labeled with DiI, a red cell membrane dye, and B16F10 cells were labeled with CFSE, a green dye that binds proteins in cells. As requested by the reviewer, we have magnified the representative eBMDMs that phagocytose B16F10 cells in Figure 3m (shown as Figure 2m in the revised manuscript). Both flow cytometry analysis (in Figure 2k and 2l) and fluorescence images (in Figure 2m) showed the phagocytic ability of macrophages after the indicated treatment. In addition, we also supplemented the quantitative analysis of Figure 2m in Figure S9.

Fig. 2 (k-l) Flow cytometry and quantitative analysis of the phagocytosis by BMDMs after the specified treatment. **(m)** Fluorescence images of the phagocytosis of BMDMs. B16F10 cells were labeled with CFSE (green), the cell membrane of eBMDM was labeled with DiI (red), and the nuclei of both cells were stained with DAPI (blue). The white arrows point to the macrophages phagocytized B16F10. Data are presented as mean \pm SD ($n = 3$). Statistical significance was calculated via one-way analysis of variance (ANOVA) with a Tukey post-hoc test. * $P < 0.5$; ** $P < 0. 1$; *** $P < 0.01$; **** $P < 0.001$.

Fig. S9 Quantitative analysis of the phagocytosis of BMDMs after the specified treatment in figure 2m.

5. Please change the use of eMac in the manuscript. There is a significant difference if the experiments were carried out with bone marrow derived macrophages or RAW cells. The latter is not translatable. If eBMDM were used, use eBMDM in the text and Figures. If engineered RAW cells were used, indicate as eRAW cells (or equivalent) in the text and Figures.

Reply: Thanks for pointing out this issue. We have corrected the abbreviation in the revised manuscript and supporting information. Whereas eBMDM and eRAW are used instead of eMac in the Results section when the particular type of macrophages is described, eMas is used in Introduction and Discussion section when we generally mention about engineered macrophages.

6. Figure 4a,b,c require biological replicates and statistics. One cannot draw any conclusion based on the luminescent pictures themselves.

Reply: Thanks for the useful comments. We have supplemented the biological replicates and statistics for Figure 4a, b and c in in the revised supporting information (Figure S10).

Fig. S10 (a) Quantitative analysis of fluorescence in figure 3a. (b) Quantitative analysis of bioluminescence in figure 3b. (c) Quantitative analysis of bioluminescence in figure 3c. Data are representative of three independent experiments with similar results. Values for n represent biologically independent samples. Data are presented as mean \pm SD (n = 3). Statistical significance was calculated via one-way ANOVA with a Tukey's multiple comparison test in c. * $P < 0.5$; ** $P < 0.1$; *** $P < 0.01$; **** $P < 0.001$. Source data are provided as a Source Data file.

7. Interpretation of data in SFig S4 is generous. It could be that the RAW cells are alive but that the lipophilic dye is degraded, or that the RAW cells have undergone multiple rounds of cell division and dilute the membrane dye. Please repeat using a genetically encoded luminescent reporter.

Reply: Thanks for the helpful comments. As requested by the reviewer, we integrated luciferase reporter into RAW264.7 by lentiviral transduction to obtain eRAW264.7 with stable luciferase expression (eRAW264.7-luci), and then eRAW264.7-luci cells were adoptively transferred and the luminescence intensity was evaluated at different time points. In agreement with our earlier results, the adoptively transferred eRAW-luci cells primarily distributed in the liver, lung and tumor, and eRAW264.7-luci could survive for about 9 days in the tumor tissue (Fig. S11b).

Fig. S11b Biodistribution and survival time of eRAW264.7 *in vivo*. The biodistribution of the eRAW264.7 cells in major organs and the tumor tissue at different time points after *i.v.* injection of luciferase-expressing eRAW264.7.

8. For Figure s6, the experimental timeline is not clear. When were tumors heated and when was the IFN γ administered?

Reply: Thank you for the useful comments. iWarm-controlled in-situ heating or the administering of IFN- γ was performed at day 1, 4, 7, 10, 12, 15, 18 and 22. And we have supplemented the illustration in Figure S13a in the revised manuscript.

Fig. S13 (a) Illustration of B16F10 tumor therapy *in vivo* with eBMDM via the remote control of locoregional hyperthermia or IFN- γ .

9. Figure 5j, k, are uninterpretable and unquantified IF images.

Reply: Thank you for the useful comments. We have magnified the representative eBMDMs in Figure 5j and 5k and arrows of the same color were used to mark the target cells (shown as Figure 4j and K in the revised manuscript). In addition, we also added the quantitative analysis of Figure 4j and k in Figure S21.

Fig. 4 (j) Multiplex IHC images of M1 macrophage and eBMDM infiltration in tumor tissues. In merged Figures, Dil-positive and CD86-positive cells (orange arrows) represent M1 eBMDM, CD86-positive only cells (green arrows) represent re-polarized TAMs.

Fig. 4 (k) IHC images of M2 macrophage infiltration in tumor tissues. The white arrows point to the M2 macrophages.

Fig. S21 Quantitative analysis of the ratio of M1 macrophages in tumor tissue (a), eBMDM in tumor tissue (b) and the ratio of M1 eBMDM in M1 macrophages (c) after the specified treatment in Figure 4j. (d) Quantitative analysis of the ratio of M2 macrophages after the specified treatment in Figure 4k. Data are representative of independent experiments with similar results. Values for n represent biologically independent samples. Data are presented as mean \pm SD (n = 3). Statistical significance was calculated via one-way ANOVA with a Tukey's multiple comparison test in d. * P < 0.5; ** P < 0.1; *** P < 0.01; **** P < 0.001. Source data are provided as a Source Data file.

10. Figure 6b is impossible to interpret. There is no validation that the CLo really depletes TAMs. Figure 6c requires biological replicates with statistics.

Reply: Thank you for highlighting this point. We have simplified the groups and showed the statistical analysis

in Figure 6b (shown as Figure 5d in the revised manuscript) to make it easier to understand. In particular, the treatment by eBMDM following heat shock and Clo treatment (eBMDM+HS+Clo) are less effective in inhibiting tumor growth, as compared with eBMDM treatment following heat shock (eBMDM+HS). We believe the difference is largely due to depletion of TAMs by clodronate liposomes. To further confirm the depletion, we also evaluated the total macrophages in melanoma tissue by flow cytometry (Figure 6e, shown as Figure 5f in the revised manuscript). It is obvious that the Clo treatment significantly depletes macrophages. The quantitative analysis of Figure 5f was showed in Figure 5g.

Fig. 5 (c-d) The inhibition of tumor growth of tumor-bearing mice after the indicated treatment. (f) Flow cytometric analysis of the macrophages infiltration in tumor tissues after the indicated treatment. CD11b⁺F4/80⁺ is the marker of macrophages. (g) The quantitative analysis of F4/80⁺CD86⁺ macrophages (M1 phenotype) in tumor tissues after the indicated treatment by flow cytometry.

11. Figure 7g does not evidence that the intestines have been brought up to 42 °C. The thermal photographs show that the surface temperature is 42 °C. The claims that inflammatory bowel disease treatment or Hirschsprung disease treatment can be addressed by this approach is not supported by the data.

Reply: Thanks for the helpful suggestion, and we agree with the reviewer’s view that the approach is not supported by the data. Therefore, we have deleted this part accordingly.

12. The claim that IFN-gamma production “was reversible and can be terminated simply through switching off the iWarm to avoid the unwanted side effects, such as cytokine release syndrome and immune effector cell-associated neurotoxicity syndrome that are commonly encountered in other cell therapies, such as CAR-T therapy” is not supported by the data. In fact, the authors show that the IFN-gamma levels do not come back down even after 72hrs after heat shock (Figure 3a,b,c) (shown as Figure 2a-c in the revised manuscript and also below).

Fig. 2 (a) Illustration of engineering process of BMDM and the polarization of eBMDM by wireless-controlled secretion of

IFN- γ . (b) The secretion of IFN- γ by eBMDM after HS at 42 °C for 30 min. (c) The increased IFN- γ by eBMDM after HS at 42 °C for 30 min at day 1 and day 3, respectively. The green arrow in (b) and (c) refers to the time point of HS.

Reply: Thanks for pointing out this issue. Since the temperature elevation contributes to IFN- γ production by activating endogenous gene, the secreted IFN- γ binds to its receptor (the heterodimeric IFNGR1/IFNGR2 receptor complex) and induces the activation of transcription elements (such as members of the signal transducer and activator of transcription (STAT) family, mainly STAT4¹, T-box transcription factor (T-bet)², activator protein 1 (AP-1)³, or Eomes⁴), which further drives the production of IFN- γ . In addition, IFN- γ may also stimulate antigen-presenting cells to secrete IL-12, which triggers the re-activation of the IFN- γ production cycle. This phenomenon is known as the positive feedback loop of IFN- γ synthesis⁵. Thus, there is a continuous increase in IFN- γ production *in vitro*. We also detected the IFN- γ levels in tumor tissues after the treatment at different time point (Fig. S20a). After eRAW264.7+iWarm treatment, we observed a significant increase of IFN- γ after HS in two days and the IFN- γ level decreased to baseline level without HS in 4 days (Fig. S20b).

Fig. S20 (a) Illustration of B16F10 tumor therapy *in vivo* with eRAW264.7 via remote control of locoregional hyperthermia. (b) IFN- γ level detected in tumor tissue after the treatment *in vivo*.

Minor points

13. Please remove subjective language throughout the paper, including but not limited to “Encouraged by these exciting results” page 6, “interestingly, emacs could migrate” page 7

Reply: Thanks for the suggestion. We have deleted those subjective language in the manuscript.

14. Please include citations of spatial / temporal control of dCas9 by light (Polstein et al. Nat. Chem. Biol. 2015), heat (Gamboa et al. ACS Chem Biol 2020), and small molecules (de Souza et al. Nat. Methods 2015).

Reply: Thanks for the suggestion. We have added the citations in the revised manuscript.

Reviewer #3

1. The authors state that eRAW264.7 cells could survive in tumor tissue for up to 10 days. How long did the eBMDMs remain in the tumor and healthy tissue? To better understand the proposed system, it would also be interesting to know the ratio of eBMDMs compared to host TAM.

Reply: Thanks for the useful comments. We integrated luciferase reporter into eBMDM by lentiviral transduction to obtain eBMDM with luciferase expression (eBMDM-luci), and then eBMDM-luci cells were adoptively transferred and the luminescence intensity was evaluated to study their distribution and residence *in vivo*. In agreement with our earlier results, the adoptively transferred eBMDM-luci primarily distributed in the liver, lung and tumor, and eBMDM-luci could reside in the tumor tissue for about 7 days, the lung for 5 days, and the liver for 5 days, as indicated by luminescence intensity (Fig. S11a).

Fig. S11a Survival time of eBMDM *in vivo*. Biodistribution of the eBMDM in major organs and the tumor tissue at different time points after *i.v.* injection of luciferase-expressing eBMDM.

Additionally, to investigate the ratio of eBMDM *in vivo*, CM-DiI-labeled eBMDM were adoptively transferred and the infiltration of eBMDM in melanoma tissue was evaluated by flow cytometry. First, we found that adoptive eBMDM accounts for about 3.8 % of the total macrophages (Fig. S17a-b). Second, we accessed the ratio of TAMs (M2 macrophage) and the ratio of eBMDM to TAMs by flow cytometry. eBMDM+HS treatment could substantially reduce the TAMs ratio as compared with other groups (Fig. S17c), and the percentage of adoptive eBMDM in TAMs increased substantially from 14.6% (eBMDM group) to 65% (eBMDM+HS group) due to the decreased TAMs (Fig. S17d). Third, we accessed the ratio of M1 macrophage in total macrophages and the ratio of M1 eBMDM in M1 macrophage by flow cytometry. The result showed that eBMDM+HS treatment could substantially induce the M1 phenotype of macrophages as compared with other groups (Fig. S17e), and the ratio of adoptive eBMDM to M1 macrophage was about ~16% (Fig. S17f). Furthermore, we found that M1 eBMDM account for ~4.9 % of the total M1 macrophages (Fig. S17g-h).

Fig. S17 (a) Flow cytometry analysis the infiltration of eBMDM in melanoma tissue after the indicated treatment. DiI+ is the marker of eBMDM. (b) Quantitative analysis the eBMDM ratio in macrophages in a. (c) Flow cytometry analysis the M2 polarization of macrophages in tumor tissues after the indicated treatment. CD206+ is the marker of TAMs (M2 macrophages). (d) Quantitative analysis the percentage of adoptive eBMDM in TAMs. (e) Flow cytometry analysis the M1 polarization of macrophages in tumor tissues after the indicated treatment. CD86+ is the marker of M1 macrophages. (f) Quantitative analysis the percentage of adoptive eBMDM in M1 macrophages. (g) Flow cytometry analysis the ratio of M1 eBMDMs in M1 macrophage after the indicated treatment. (h) Quantitative analysis the M1 eBMDM ratio in M1 macrophages in g. Data are representative of independent experiments with similar results. Values for n represent biologically independent samples. Data are presented as mean \pm SD (n = 3). Statistical significance was calculated via one-way ANOVA with a Tukey's multiple comparison test in d. * $P < 0.5$; ** $P < 0.1$; *** $P < 0.01$; **** $P < 0.00$. Source data are provided as a Source Data file.

2. In fig 2, induction of IFN γ is measured at 37°C, 42°C, 45°C and 48°C. What level of induction is seen between 37°C-42°C?

Reply: Thank you for highlighting this point. Figure 2 (shown as Figure 1 in the revised manuscript) presents the induction of EGFP expression instead of IFN- γ production by thermo-activation of EGFP gene. Since the optimal temperature of heat-shock promoter is about 42°C, the induction level between 37°C-42°C should be 2.4%-100%.

3. Throughout the manuscript, the reader gets the impression that heat inducible expression of IFN γ in macrophages is safe although no data on toxicity are presented. Given the infiltration of eBMDMs in normal tissue as well, what would be the effect of slightly elevated body temperatures? This should be tested in an appropriate model (e.g. experimental endotoxemia model)

Reply: Thanks for the comment. Actually, the data on toxicity has been presented in Figure S15 and S16. After eMac*s*+iWarm treatment, there is no toxicity to the major organs, as demonstrated by H&E staining (Fig. S15), and hematological evaluation suggested that the treatment neither caused any damage to the liver and kidney, nor resulted in any inflammation (Fig. S16), suggesting the safety of such a treatment modality. We also detected the IFN- γ level in tumor tissues and serum after the treatment. After eBMDM+HS treatment, though IFN- γ level was found significantly increased in tumor tissue compared with other groups, the level in serum remains unchanged (Fig. S19).

Fig. S12 H&E staining of major organs at different time points after the indicated treatment.

Fig. S13 Hematological evaluation after the indicated treatment. The counts of (a) white blood cell, (b) lymphocyte, (c) monocyte, (d) AST (aspartate aminotransferase), (e) ALT (alanine aminotransferase), (f) AST/ALT, (g) total protein, (h) albumin, (i) globulin, (j) albumin/globulin, (k) blood urea nitrogen, (l) uric acid in serum at day 21. Data represent mean \pm SD ($n = 3$). Statistical significance was calculated via one-way analysis of variance (ANOVA) with a Tukey post-hoc test. * $P < 0.5$; ** $P < 0.1$; *** $P < 0.01$; **** $P < 0.001$.

IFN- γ levels in tumor tissues (**Fig. 4h**) and serum (**Fig. S19**) collected from mice after the indicated treatment. Data are representative of independent experiments with similar results. Values for n represent biologically independent samples. Data are presented as mean \pm SD ($n = 3$). Statistical significance was calculated via one-way ANOVA with a Tukey's multiple comparison test in d. * $P < 0.5$; ** $P < 0.1$; *** $P < 0.01$; **** $P < 0.001$. Source data are provided as a Source Data file.

4. In Figure 5 the authors claim that heat inducible expression of IFN γ drives an increase in CD4 and CD8 T cells. However, as the data only shows increase in CD4 $^{+}$ and CD8 $^{+}$ cells out of CD3 $^{+}$ T cells, it is not possible to conclude that T cells are increased. Instead, the authors should as a minimum show the frequency of CD4 and CD8 T cells out of live cells or preferentially, number of cells per tissue weight. Also, in control mice,

less than 40% of the CD3⁺ cells are CD4 or CD8, which makes the reader question the gating strategy used. As a minimum, this should be available in the supporting information.

Reply: Thanks for pointing out this issue. The Figure 5e-g (shown as Figure 4e-g in the revised manuscript) showed the CD4⁺ cells and CD8⁺ cells within (not without) the CD3⁺ cells. So, we suppose that heat inducible expression of IFN- γ drives an increase in the ratio of CD4⁺ and CD8⁺ T cells. We have corrected the description regarding CD4⁺ and CD8⁺ T cells. In addition, the gating strategy were available in Figure S23.

Fig. S23 Gating strategy of CD4⁺ and CD8⁺ T cells within CD3⁺ T cells.

5. In Figure 6, the authors use clodronate liposomes to prove the effect of eBMDMs. However, clodronate Liposomes depletes all macrophages and the differences between clodronate treated eBMDMs + HS and eBMDMs + HS seem minor. To prove that the observed treatment effect is due heat inducible IFN γ expression, it would make more sense to include an IFN γ /IFN γ R blocking antibody.

Reply: Thanks for the useful comments. Actually, in Figure 6e and 6f (shown as Figure 5f-g in the revised manuscript), our results showed that there is significant difference between the eBMDMs+HS group and eBMDMs+HS+Clo group ($P < 0.001$), which suggests the Clo treatment significantly depleted macrophages.

Fig. 5 (f) Flow cytometric analysis the macrophages infiltration in tumor tissues after the indicated treatment. CD11b⁺F4/80⁺ is the marker of macrophages. (g) The quantitative analysis of F4/80⁺ CD11b⁺ macrophages in tumor tissues after the indicated treatment by flow cytometry. Data are presented as mean \pm SD (n = 3). Statistical significance was calculated via one-way analysis of variance (ANOVA) with a Tukey post-hoc test. * $P < 0.5$; ** $P < 0.1$; *** $P < 0.01$; **** $P < 0.001$.

According to the suggestion of the reviewer, to prove that the observed treatment effect is due to heat inducible IFN- γ expression, we evaluated the treatment effectiveness over the IFN- γ -blocking melanoma models, in which anti-mouse IFN- γ antibody was used for the neutralization of IFN- γ *in vivo* (Fig. 6a). First, we found that anti-IFN- γ treatment attenuated the antitumor activity of eBMDM+HS treatment (Fig. 6b and c). Next, we observed IFN- γ level decreased in the melanoma tissues after anti-IFN- γ treatment compared with control group treated with isotype control antibody, and IFN- γ level also decreased in eBMDM+HS+anti-IFN- γ group compared with eBMDM+HS group (Fig. 6d). Then, we accessed the infiltration of M1 and M2 macrophages

in melanoma tissue by flow cytometry. The ratio of M1 macrophages significantly decreased while the ratio of M2 macrophages significantly increased after eBMDM+HS treatment in the presence of anti-IFN- γ , as compared with the treatment by eBMDM+HS (Fig. 6e-h). Furthermore, the infiltration of CD8⁺ T cells in the melanoma tissue was detected by flow cytometry. The ratio of CD8⁺ T cells significantly decreased in the eBMDM+HS+anti-IFN- γ group, as compared with eBMDM+HS group (Fig. 6i and j). Collectively, these results demonstrate that the effectiveness of eBMDMs for the treatment of melanoma under the iWarm-enabled locoregional hyperthermia is due to heat inducible IFN- γ expression.

Fig. 6 IFN- γ neutralization attenuates antitumor activity of adoptive eBMDM therapy mediated by wireless-controlled iWarm. (a) Illustration of IFN- γ neutralization and adoptive eBMDM therapy with wireless iWarm in vivo. (b) In vivo bioluminescence images of mice after the specified treatment at day 11. (c) The inhibition of tumor growth after the indicated treatment. (d) IFN- γ levels in tumor tissues collected from mice after the indicated treatment. Flow cytometry analysis the M1 polarization (e) and the M2 polarization (g) of macrophages in tumor tissues after the indicated treatment. CD86⁺ is the marker of M1 macrophages, while CD206⁺ is the marker of M2 macrophages. Quantitative analysis of the M1 (f) and M2 (h) ratio of macrophages in e and g. (e) Flow cytometry analysis the CD8⁺ T cells in tumor tissues after the indicated treatment. (j) Quantitative analysis of the ratio of CD8⁺ T cells. Data are representative of three independent experiments with similar results. Values for n represent biologically independent samples. Data are presented as mean \pm SD (n = 3). Statistical significance was calculated via one-way ANOVA with a Tukey's multiple comparison tests in d, f, h and j. * P < 0.5; ** P < 0.1; *** P < 0.01; **** P < 0.001. Source data are provided as a Source Data file.

References

1. Thieu VT, *et al.* Signal transducer and activator of transcription 4 is required for the transcription factor T-bet to promote T helper 1 cell-fate determination. *Immunity* **29**, 679-690 (2008).
2. Kanhere A, *et al.* T-bet and GATA3 orchestrate Th1 and Th2 differentiation through lineage-specific targeting of distal regulatory elements. *Nature communications* **3**, 1268 (2012).
3. Negishi H, Taniguchi T, Yanai H. The Interferon (IFN) Class of Cytokines and the IFN Regulatory Factor (IRF) Transcription Factor Family. *Cold Spring Harb Perspect Biol* **10**, (2018).
4. Pearce EL, *et al.* Control of effector CD8⁺ T cell function by the transcription factor Eomesodermin. *Science* **302**, 1041-1043 (2003).
5. Garris CS, *et al.* Successful Anti-PD-1 Cancer Immunotherapy Requires T Cell-Dendritic Cell Crosstalk Involving the Cytokines IFN- γ and IL-12. *Immunity* **49**, 1148-1161.e1147 (2018).
6. Veillette A, Chen J. SIRP α -CD47 Immune Checkpoint Blockade in Anticancer Therapy. *Trends Immunol* **39**, 173-184 (2018).
7. Feng M, Jiang W, Kim BYS, Zhang CC, Fu YX, Weissman IL. Phagocytosis checkpoints as new targets for cancer immunotherapy. *Nat Rev Cancer* **19**, 568-586 (2019).

REVIEWER COMMENTS

Reviewer #2 (Remarks to the Author):

The authors were largely responsive to the concerns raised by this reviewer. Several concerns remain.

- The response that "...the expression of IFN- γ driven by heat-shock promoter was not transient due to the integration of transgene into the host genome by lentiviral transduction, thereby making eBMDM longer periods of transgene expression..." is not factually accurate. Heat-shock promoters exhibit negligible activity body temperature. If the heat stimulus is removed, transgene expression should no longer occur. If the authors want to claim that IFN-g production is no longer transient after the heat-shock circuit is integrated into the host genome (which is contrary to the field), then they would need to provide data to support this hypothesis. Also, a non-transient system directly opposes the goal of the study to design a turn ON transient system.
- Figure 2b and 2c are contradictory. In figure 2B, IFN-g levels continue to rise at 48 hrs, yet in figure 2c, it drops by day 2. If IFN-g triggers a positive feedback loop as described by the authors, then Figure 2c is not consistent with that mechanism. If IFN-g does not trigger a positive feedback loop, then Figure 2b is not consistent.
- The text describing Figure 1e is ambiguous and unprecise. "... we first integrated the genome of RAW264.7 cells with a plasmid encoding EGFP (enhanced green fluorescence protein) reporter driven by HSP70 promoter through the lentiviral transfection (Fig. 1d)." Was this experiment performed by lentiviral transduction? Or did they transfect the cells with lentiviral plasmids? Is it not clear. The captions and figure also state that the cells were transfected.
- The abstract language "innovative paradigm" is subjective.
- The language "Inspired by the booming development of artificial intelligence devices" is not relevant as the device itself is not operating under AI.

Reviewer #3 (Remarks to the Author):

To the authors, thank you for a very thorough rebuttal letter. In general, the authors have addressed most of my original concerns. However, a few minor comments remains.

In figure S17 b,d,f,h the authors use the term macrophages, TAMs, M1 where macrophages are gated as F4/80+, CD11b+, TAMs as CD206+ and M1 as CD86+. For clarity, as all macrophages are tumor-associated, it would be much better to name F4/80+, CD11b+ as TAM, CD206+ cells as M2-like TAM and CD86+ cells as M1-like TAM.

In figure S23 the authors show that gating strategy of CD4 and CD8 T cells. In the manuscript (p. 9 | 18-19) the authors write: "and the substantial increase in the ratio of CD4+ and CD8+ T cells was also detected in eBMDM+HS or eRAW264.7+HS group (Fig. 4e-g and Fig. S18e-g)".

However, its only the ratio of CD8 and CD4 positive cells out of CD3 positive cells that is calculated. Since an increase in CD4 and CD8 ratio could result from both an increase in these cells but also a decrease in CD3+, CD4-, CD8- cells, it would be much more valuable to calculate the freq of CD8+ and CD4+ T cells from live cells or CD45+ cells.

In figure 6 the authors have tested the effect of an IFNg blocking antibody and whereas the effect is clear on macrophages and T cells (panel e-j), the effect on tumor growth in panel b-c is less obvious. It would more convincing if growth curves were combined as in Fig.5d and differences was backup by a statistical analysis

Reviewer #4 (Remarks to the Author):

The paper is very interesting and provides a valuable strategy for adoptive transfer of genetically engineered macrophage in order to switch, by remotely controlled heat activation, the polarization of tumor associated macrophages in melanoma. A number of major concerns have been raised by the reviewers that were correctly addressed in the revised version, with additional experiments performed and careful and thorough responses provided. So I consider that the revised manuscript supports the conclusion and claims, with relevant references and sound methodology and deserves publication.

Point-by-Point Response to the Reviewers' Comments

(Manuscript ID: NCOMMS-22-52387B)

Reviewer #2

1. The response that "...the expression of IFN- γ driven by heat-shock promoter was not transient due to the integration of transgene into the host genome by lentiviral transduction, thereby making eBMDM longer periods of transgene expression..." is not factually accurate. Heat-shock promoters exhibit negligible activity body temperature. If the heat stimulus is removed, transgene expression should no longer occur. If the authors want to claim that IFN-g production is no longer transient after the heat-shock circuit is integrated into the host genome (which is contrary to the field), then they would need to provide data to support this hypothesis. Also, a non-transient system directly opposes the goal of the study to design a turn ON transient system.

Reply: Thanks for pointing out this issue. We agree with the reviewer that heat-shock promoters exhibit negligible activity at body temperature and if the heat stimulus is removed, transgene expression should no longer occur. We are sorry about the unclear description of "...the expression of IFN- γ driven by heat-shock promoter was not transient due to the integration of transgene into the host genome by lentiviral transduction, thereby making eBMDM longer periods of transgene expression...". What we expect to convey is as follows: The integration of transgene containing a heat-shock promoter into host genome by integrating lentiviral transduction provides the possibility of stable, repeatable induction of protein expression upon the exposure of eBMDM to heat stimulus. In contrast, the transfection by non-viral vectors, which are incapable of integrating the transgene into the host genome, is unable to afford the stable, repeatable, heat-inducible protein expression over a relatively long period of time.

2. Figure 2b and 2c are contradictory. In figure 2B, IFN-g levels continue to rise at 48 hrs, yet in figure 2c, it drops by day 2. If IFN-g triggers a positive feedback loop as described by the authors, then Figure 2c is not consistent with that mechanism. If IFN-g does not trigger a positive feedback loop, then Figure 2b is not consistent.

Reply: Thanks for the comment. It should be noted that the Y axis is the accumulative amount of IFN- γ (shown as concentration of IFN- γ , pg/ml) in Figure 2b, whereas the Y axis in Figure 2C is the increment of IFN- γ (pg/ml), which is calculated as $Y = (\text{the concentration of IFN-}\gamma \text{ in day } x) - (\text{the concentration of IFN-}\gamma \text{ in day } x-1)$.

Fig. 2 (b) The secretion of IFN- γ by eBMDM after HS at 42 °C for 30 min. (c) The increased IFN- γ by eBMDM after HS at 42 °C for 30 min at day 1 and day 3, respectively. The green arrow in (b) and (c) refers to the time point of HS.

3. The text describing Figure 1e is ambiguous and unprecise. "... we first integrated the genome of RAW264.7 cells with a plasmid encoding EGFP (enhanced green fluorescence protein) reporter driven by HSP70 promoter through the lentiviral transfection (Fig. 1d)." Was this experiment performed by lentiviral transduction? Or did they transfect the cells with lentiviral plasmids? Is it not clear. The captions and figure also state that the cells were transfected.

Reply: Thanks for pointing out this issue. We are sorry about that the unprecise description. The experiments in Figure 1d-h were performed by transfection of eGFP plasmids using commercial Lipofectamine, and the experiments in Figure 1i-j were performed by lentiviral transduction. We have corrected the relevant description in the revised manuscript, Figure 1i and also below.

“... we first transfected RAW264.7 cells with a plasmid encoding EGFP (enhanced green fluorescence protein) reporter driven by HSP70 promoter using Lipofectamine (Fig. 1d).”

Fig. 1 (i) Illumination time-dependent HS-mediated ON-OFF kinetics of transgene expression. By lentiviral transduction, EGFP expression was monitored everyday after the heat shock at 42 °C for 30 min, which was carried at day 0, day 3 and day 6, respectively.

4. The abstract language “innovative paradigm” is subjective.

Reply: Thanks for the suggestion. After our second thought, we have changed the “innovative paradigm” to “creative strategy”.

5. The language “Inspired by the booming development of artificial intelligence devices” is not relevant as the device itself is not operating under AI.

Reply: Thanks for the suggestion. We have deleted the language “Inspired by the booming development of artificial intelligence devices”.

Reviewer #3

1. In figure S17 b,d,f,h the authors use the term macrophages, TAMs, M1 where macrophages are gated as F4/80+, CD11b+, TAMs as CD206+ and M1 as CD86+. For clarity, as all macrophages are tumor-associated, it would be much better to name F4/80+, CD11b+ as TAM, CD206+ cells as M2-like TAM and CD86+ cells as M1-like TAM.

Reply: Thanks for the insightful comments. We have corrected the relevant description in revised manuscript (highlighted as yellow below) and Figure S17 b,d,f,h.

First, we found that adoptive eBMDM accounts for ~ 3.8 % of the total macrophages (TAMs) (Fig. S17a-b). As compared with other groups, eBMDM+HS or eRAW264.7+HS treatment could substantially induce the M1 phenotype of TAMs, with the observable upregulation of M1 markers (*Ifng*, *Cd86*, *Il6*, *Ccl2* and *Tnf*) (Fig. 4a and Fig. S18a) and downregulation of M2 markers (*Cd206*, *Il10*, *Arg1*, *Fizz1*) (Fig. 4b and Fig. S18b) as well as increased M1/M2 ratio (Fig. 4c-d and Fig. S18c-d). Flow cytometry analysis showed that eBMDM+HS treatment could substantially reduce the number of M2-like TAMs as compared with other groups (Fig. S17c), and the percentage of adoptive eBMDM in M2-like TAMs increased substantially from 14.6 % (eBMDM group) to 65% (eBMDM+HS group) due to the decreased M2-like TAMs (Fig. S17d). Then, we assessed the ratio of M1-like TAMs and the ratio of M1 eBMDM in M1-like TAMs population by flow cytometry. The result showed that eBMDM+HS treatment could substantially induce the M1 phenotype of TAMs as compared with other groups (Fig. S17e), and the ratio of adoptive eBMDM to M1-like TAMs accounts for ~16% in eBMDM+HS group (Fig. S17f). Also, we found that M1 eBMDM

accounts for ~ 4.9 % of the total population of M1-like TAMs (Fig. S17g-h), and the substantial increase in the ratio of CD8⁺ T cells was also detected in eBMDM+HS or eRAW264.7+HS group (Fig. 4e-g and Fig. S18e-g).

Fig. S17 (a) Flow cytometry analysis the infiltration of eBMDM in melanoma tissue after the indicated treatment. DiI⁺ is the marker of eBMDM. (b) Quantitative analysis the eBMDM ratio in TAMs in a. (c) Flow cytometry analysis M2-like TAMs in tumor tissues after the indicated treatment. CD206⁺ is the marker of M2-like TAMs. (d) Quantitative analysis the percentage of adoptive eBMDM in TAMs. (e) Flow cytometry analysis the M1-like TAMs in tumor tissues after the indicated treatment. CD86⁺ is the marker of M1-like TAMs. (f) Quantitative analysis the percentage of adoptive eBMDM in M1-like TAMs. (g) Flow cytometry analysis the ratio of M1 eBMDMs in M1-like TAMs after the indicated treatment. (h) Quantitative analysis the M1 eBMDM ratio in M1-like TAMs in g. Statistical significance was calculated via one-way analysis of variance (ANOVA) with a Tukey post-hoc test. **P* < 0.5; ***P* < 0.1; ****P* < 0.01; *****P* < 0.001.

- In figure S23 the authors show that gating strategy of CD4 and CD8 T cells. In the manuscript (p. 91 18-19) the authors write: “and the substantial increase in the ratio of CD4⁺ and CD8⁺ T cells was also detected in eBMDM+HS or eRAW264.7+HS group (Fig. 4e-g and Fig. S18e-g)”. However, it is only the ratio of CD8 and CD4 positive cells out of CD3 positive cells that is calculated. Since an increase in CD4 and CD8 ratio could result from both an increase in these cells but also a decrease in CD3⁺, CD4⁻, CD8⁻ cells, it would be much more valuable to calculate the freq of CD8⁺ and CD4⁺ T cells from live cells or CD45⁺ cells.

Reply: Thanks for the helpful comments. As requested by the Reviewer, we have added the frequency of CD8⁺ T cells from CD45⁺ cells in Figure 4e-f (eBMDM treatment) and Figure S18e-f (RAW264.7 treatment). The gating strategy is shown in Figure S23 in the revised manuscript, which is also shown below.

Fig. 4 (e) Flow cytometry analysis the CD8⁺ T cells in tumor tissues after eBMDM treatment with or without heat shock. (f) Quantitative analysis of the ratio of CD8⁺ T cells in CD45⁺ cells.

Fig. S18 (e) Flow cytometry analysis the CD8⁺ T cells in tumor tissues after eRAW264.7 treatment with or without heat shock. (f) Quantitative analysis of the ratio of CD8⁺ T cells in CD45⁺ cells.

Fig. S23 Gating strategy of CD8⁺ T cells within CD45⁺ cells or CD3⁺ T cells and gating strategy of M1 and M2 macrophages.

3. In figure 6 the authors have tested the effect of an IFN γ blocking antibody and whereas the effect is clear on macrophages and T cells (panel e-j), the effect on tumor growth in panel b-c is less obvious. It would more convincing if growth curves were combined as in Fig.5d and differences were backed up by a statistical analysis.

Reply: Thanks for the useful comment. We have added the statistical analysis of the growth curves. As shown in Figure 6c in revised manuscript and also below, statistical analysis showed that anti-IFN- γ

eBMDMs+HS treatment attenuated the antitumor activity of eBMDMs+HS treatment ($*P < 0.5$) (Fig. 6b-d).

Fig. 6 (b) *In vivo* bioluminescence images of mice after the specified treatment at day 11. Average (c) and individual (d) tumor growth curves after the specified treatment. Data are presented as mean \pm SD ($n = 3$). Statistical significance was calculated via two-tailed t test in c. $*P < 0.5$; $**P < 0.1$; $***P < 0.01$; $****P < 0.001$.

Reviewer #4

1. The paper is very interesting and provides a valuable strategy for adoptive transfer of genetically engineered macrophage in order to switch, by remotely controlled heat activation, the polarization of tumor associated macrophages in melanoma. A number of major concerns have been raised by the reviewers that were correctly addressed in the revised version, with additional experiments performed and careful and thorough responses provided. So I consider that the revised manuscript supports the conclusion and claims, with relevant references and sound methodology and deserves publication.

Reply: Thank you for the useful comment and recommendation.

REVIEWERS' COMMENTS

Reviewer #2 (Remarks to the Author):

The authors have addressed all my remaining concerns, except that the proposed new language in the abstract to self-describe the approach as "creative strategy" is still subjective. I'll leave this to editor discretion to keep or not.

Reviewer #3 (Remarks to the Author):

Thank you for addressing all remaining concerns. I do not have any further concerns and would like to take the opportunity for congratulating the authors on the very nice work